# FEAST: A flow cytometry-based toolkit for interrogating microglial engulfment of synaptic and myelin proteins

Lasse Dissing-Olesen[1,2,3,7], Alec J. Walker[1,2,3,7], Qian Feng[4], Helena J. Barr[1,2,3], Alicia C. Walker[1,2], Lili Xie[4], Daniel K. Wilton[1,2,3], Indrani Das[1,2], Larry I. Benowitz [4,5] & Beth Stevens [1,2,3,6] ✉

Although engulfment is a hallmark of microglia function, fully validated platforms that facilitate high-throughput quantification of this process are lacking. Here, we present FEAST (*Flow cytometric Engulfment Assay for Specific Target proteins*), which enables interrogation of in vivo engulfment of synaptic material by brain resident macrophages at single-cell resolution. We optimize FEAST for two different analyses: quantification of fluorescent material inside live cells and of engulfed endogenous proteins within fixed cells. To overcome false-positive engulfment signals, we introduce an approach suitable for interrogating engulfment in microglia from perfusion-fixed tissue. As a proof-of-concept for the specificity and versatility of FEAST, we examine the engulfment of synaptic proteins after optic nerve crush and of myelin in two mouse models of demyelination (treatment with cuprizone and injections of lysolecithin). We find that microglia, but not brain-border associated macrophages, engulf in these contexts. Our work underscores how FEAST can be utilized to gain critical insight into functional neuro-immune interactions that shape development, homeostasis, and disease.

Microglia engulf synaptic material during development and contribute to aberrant synaptic loss in several models of neurodegenerative diseases[1–4]. These findings have sparked extensive interest in the pathways and receptors involved in engulfment and neuro-immune interactions in the central nervous system (CNS). Emerging questions include how microglia functionally interact with neurons in diverse paradigms, including those affecting peripheral immune signaling, host-environment interactions, disease-relevant genetic modulations, and sleep/wake cycles.

To date, most studies have assessed in vivo engulfment by microglia via immunohistochemistry and microscopy-based analyses of fixed tissue[5–7]. This approach requires laborious surface-rendering of each microglial cell through z-stacks of confocal laser-scanning microscopy images, severely limiting the number of microglia that can be analyzed. The same holds true when electron microscopy has been applied to analyze inclusions that contain synaptic material inside microglia[8,9]. We aimed to expand the scale and versatility of microglia engulfment analyses, and therefore developed a high-throughput flow cytometry-based platform referred to as Flow cytometric Engulfment Assay for Specific Target proteins (FEAST).

The basic principle of FEAST relies on the physical isolation of microglia from brain tissue, allowing the separation of microglia from non-engulfed neuronal material. Microglia are then interrogated by flow cytometry to quantify engulfed neuronal material at single-cell

¹Department of Neurology, Boston Children's Hospital, F.M. Kirby Neurobiology Center, Boston, MA 02115, USA. ²Harvard Medical School, Boston, MA 02115, USA. ³Stanley Center for Psychiatric Research, Broad Institute of MIT and Harvard, Cambridge, MA 02139, USA. ⁴Department of Neurosurgery, Boston Children's Hospital, F.M. Kirby Neurobiology Center, Boston, MA 02115, USA. ⁵Ophthalmology, Harvard Medical School, Boston, MA 02115, USA. ⁶Howard Hughes Medical Institute, Boston Children's Hospital, Boston, MA 02115, USA. ⁷These authors contributed equally: Lasse Dissing-Olesen, Alec J. Walker. ✉e-mail: beth.stevens@childrens.harvard.edu

resolution. In contrast to imaging of microglial engulfment in fixed tissue, this approach allows for rapid and unbiased analyses of microglia from the whole brain and is sensitive enough to facilitate the analysis of thousands of microglia from micro-dissected regions.

Flow cytometry has been employed previously to address questions about microglial engulfment[10–15]. However, none of these studies provide a clear strategy for overcoming pitfalls such as distinguishing material engulfed in vivo from false-positive signals arising from ex vivo engulfment or debris adherence. Here, we provide a detailed methodology designed to overcome these challenges and highlight essential considerations about technical details and important controls. Novel features include different approaches for preparing single-cell suspensions, strategies for removal of debris and prevention of ex vivo engulfment, a platform for selection of antibodies, gating strategies, and a demonstration of how FEAST can be utilized to address exploratory questions in different paradigms.

## Results

### Microglial engulfment in the optic nerve crush paradigm
To determine whether flow cytometry could be applied to accurately quantify neuronal material engulfed by microglia in vivo, we chose the optic nerve crush (ONC) paradigm. ONC has previously been shown to evoke robust engulfment by microglia in the lateral geniculate nucleus (LGN) of the visual thalamus[11]. We first labeled retinal ganglion cell (RGC) inputs, in WT mice, with Alexa Fluor (AF488) conjugated to the cholera toxin B subunit (CTB) and performed bilateral ONC to evaluate microglial engulfment of AF488 by a standard IHC approach (Fig. 1a). Inclusions of AF488 were prevalent within microglia in the LGN three days after ONC. We also observed AF488+ inclusions in microglia after isolation by Dounce homogenization and immunocytochemistry (Fig. 1b), indicating that engulfed material is maintained within microglia during the isolation process that is required for downstream flow cytometry applications.

Next, we labeled inputs from each eye with a different Alexa Fluor dye (AF488 and AF594), performed unilateral or bilateral ONC, and pooled brain regions (the LGNs and the superior colliculi, SC) receiving crushed and intact inputs for microglia isolation and downstream analysis (Fig. 1c). After unilateral ONC, the percentage of microglia containing Alexa Fluor signal derived from crushed inputs increased whereas the percentage of microglia containing Alexa Fluor signal derived from intact inputs remained similar to control mice. Additionally, when both optic nerves were crushed, individual microglia predominantly engulfed inputs from one eye or the other (Fig. 1d, e), presumably corresponding to the physically proximal inputs to which they were exposed in vivo. This experiment demonstrates that FEAST is highly sensitive for detecting engulfed neuronal material and emphasizes that false-positive signal due to the acquisition of material ex vivo (e.g., contamination with debris from distal inputs during the isolation process) is negligible in this paradigm.

### Engulfment of different fluorescent proteins
Given our ability to detect fluorescent dyes inside of microglia by FEAST, we next examined microglial engulfment of genetically encoded fluorescent proteins (FPs). To identify FPs suitable for interrogating engulfment in live microglia, we expressed FPs with different pH stabilities and sensitivities to lysosomal degradation (Fig. 2a) in RGCs using the CHX10-Cre driver line[16] crossed to commercially available Rosa-lox-stop-lox reporter lines and performed ONC.

As expected, we observed a striking inverse correlation between the detected mean fluorescent intensity (MFI) after ONC and the pKa of the engulfed fluorescent protein. The fold change in MFI between ONC and no crush for engulfed ZsGreen (pKa ~3) was comparable to the fold change in MFI for engulfed AlexaFluor dyes (Fig. 2b). In contrast, the MFI of engulfed EGFP (pKa ~6) and EYFP (pKa ~ 7) did not increase significantly after ONC (Fig. 2d, e). ONC-evoked engulfment of

TdTomato (pKa ~4.8), which has a pKa that falls within the range of the theoretical lysosomal pH (4.5–5.5), was significantly higher than no crush (Fig. 2c) but not to the same degree as ZsGreen. These data indicate that FPs with a pKa within the range of or lower than the lysosomal pH are required to detect engulfment with FEAST.

### Engulfment screen for antibodies against synaptic and myelin proteins
Next, we sought to expand FEAST to allow interrogation of microglial engulfment of endogenous synaptic proteins with an antibody-based approach. To identify antibodies suitable for labeling engulfed synaptic proteins, we performed an extensive in vitro screen. We cultured the stable microglial cell line, EOC20, and exposed one group to synaptic material in the form of synaptosomes[17] while another group was kept untreated. EOC20 cells from both groups were fixed, permeabilized, and stained with one of 42 antibodies targeting various pre- and post-synaptic proteins (Fig. 3a, b, Supplementary Table 1). We identified eight antibodies that each produced a fluorescent signal that was more than three-fold higher in cells that had been fed synaptosomes compared to unfed control cells (Fig. 3c).

Two antibodies, one against SNAP-25 and one against synapsin 1 (SYN1), fulfilled all our criteria for developing an in vivo engulfment assay. Both antibodies target highly abundant synaptic proteins and are monoclonal, thus eliminating lot-to-lot variability. Additionally, SNAP-25 is highly expressed in RGCs and could therefore be validated in the ONC paradigm[18]. SYN1-deficient mice are also available and provide an ideal control for assessing false-positive signal. We titrated these antibodies in primary microglia cultures fed WT or SYN1-KO synaptosomes, confirming their suitability for FEAST (Supplementary Fig 1).

To expand the framework of FEAST beyond the engulfment of synaptic proteins, we also tested antibodies against the myelin-associated protein, myelin basic protein (MBP) after exposing EOC20 cells to myelin debris (Fig. 3d, e, Supplementary Table 1). Three different clones produced greater than a three-fold change in fluorescent intensity above isotype controls. Clone P82H9 produced greater than a 45-fold change in cells fed myelin versus controls (Fig. 3e), supporting its utility for quantification of myelin engulfment in vivo.

### Assessing different isolation protocols and their potential pitfalls
Live microglial cells can be successfully isolated by mechanical (Dounce) homogenization of minced tissue at 4 °C as we did after ONC (Figs. 1, 2) and as we and others have done previously[19–21]. However, enzymatic digestion is required to efficiently isolate other tissue-resident macrophages such as brain-border associated macrophages (BAMs), which may play important and complementary roles in debris clearance or antigen presentation in different disease contexts. We, therefore, tested several different tissue dissociation protocols (Dounce homogenization, collagenase IV (Col-IV) enzymatic digestion, and digestion with a cold-active protease from *Bacillus licheniformis*) with the goal of assessing our ability to detect engulfment of endogenous synaptic material in microglia and BAMs isolated by each. We anticipated that each protocol might encounter different pitfalls related to false-positive signals. For example, digestion with Col-IV requires incubation at 37 °C, which presumably allows live cells to engulf synaptic material ex vivo. Additionally, synaptic debris generated by different dissociation approaches might adhere to the cells, resulting in a false-positive signal indistinguishable from material engulfed in vivo.

To minimize the possibility for ex vivo engulfment, we optimized a cocktail of inhibitors designed to block phagocytosis, endocytosis (clathrin-, caveolin-dependent and independent), and macropinocytosis by application of Dynasore, Pitstop 2, Wortmannin, and Cytochalasin D. We also included Bafilomycin A1, which inhibits

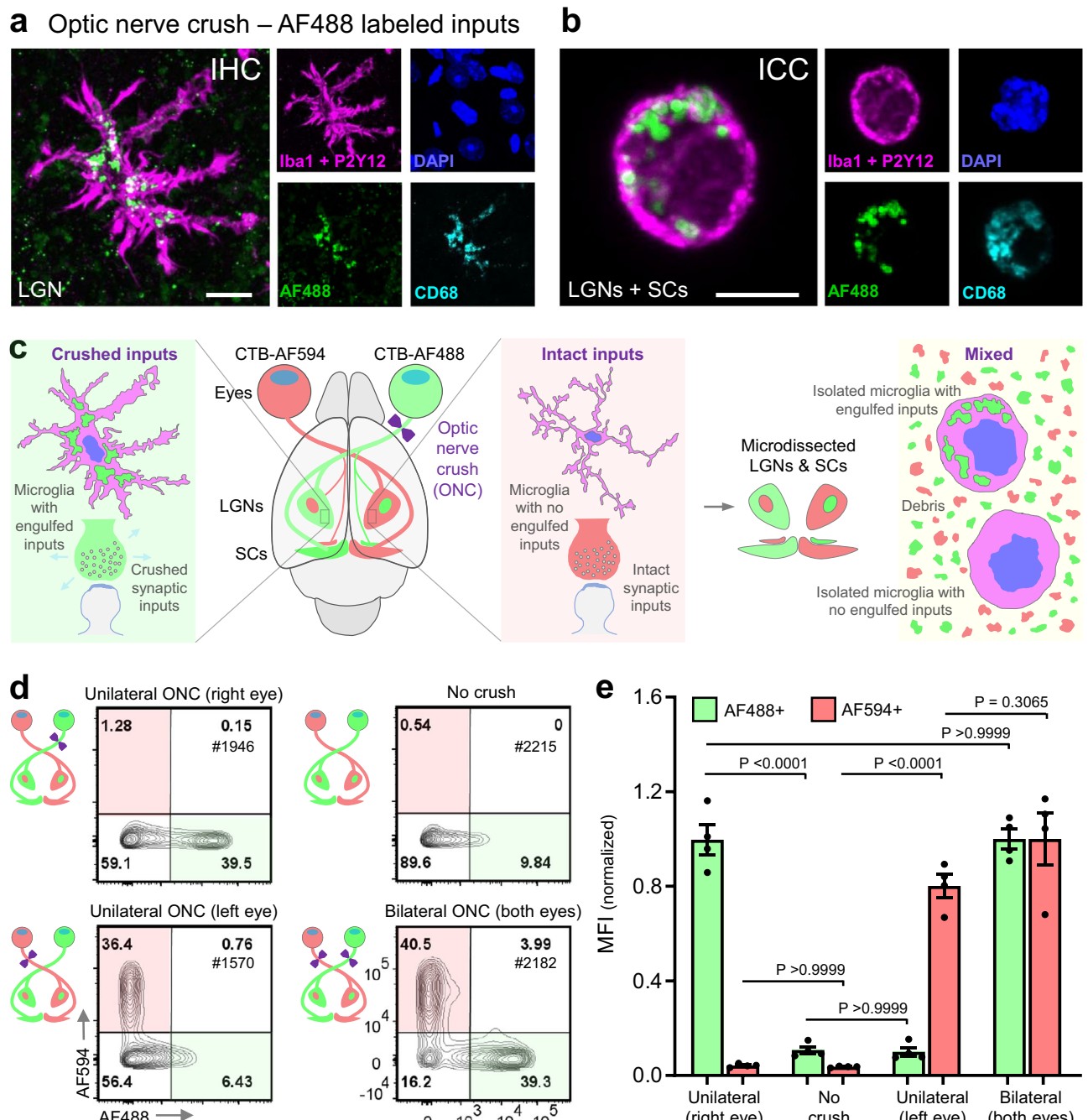

**Fig. 1 | Microglia engulfment of fluorescently labeled synaptic inputs following optic nerve crush. a** Immunohistochemistry of a microglial cell in the lateral geniculate nucleus (LGN) following bilateral optic nerve crush (ONC) of Alexa Fluor 488 (AF488)-labeled retinal ganglion cell (RGC) inputs. Microglia - IBA-1 and P2RY12, magenta; nuclei - DAPI, blue; and lysosomes - CD68, cyan. Representative image of hundreds of microglia examined. Scale bar represents 10 μm. **b** Immunocytochemistry of a microglial cell isolated from micro-dissected LGNs and superior colliculi (SCs) following ONC of AF488-labeled RGC inputs. Representative image of hundreds of microglia examined. Scale bar represents 5 μm. **c** Schematic of RGC labeling and selective engulfment of crushed inputs following unilateral ONC. RGCs in each eye were labeled by intravitreal injection of cholera toxin subunit-B conjugated with either AF594 (CTB-594) or AF488 (CTB-AF488), resulting in robust labeling of their projections to the LGN and SC. The inputs from either the right or left eye were then crushed, and microglia were isolated from

micro-dissected, pooled LGNs and SCs for quantification of engulfed AF488 and AF594 by flow cytometry. **d** Representative flow cytometry plots of microglia isolated from LGNs/SCs following unilateral ONC of either the inputs from the right (AF488-labeled) or the left (AF594-labeled) eye when no crush was conducted or following bilateral ONC. Plots are representative of 4 mice per condition quantified in (**e**). **e** Mean fluorescent intensity (MFI) of AF488 and AF594 contained in microglia from ONC or No crush conditions. Microglia were isolated from the LGNs/SCs by Dounce homogenization at 4 °C, enriched by Percoll centrifugation 3 days after ONC, and gated on the following markers: live, single cells, CD45+, CD11b+, CX3CR1+, CD206-. MFIs for AF488 or AF594 are normalized to the average MFI value for bilateral ONC for each respective color. Error bars indicate standard error of the mean. $n$ = 4 mice per condition (two females and two males, except for No crush, four females). Statistical analysis: One-way ANOVA, Bonferroni's multiple comparisons test. Source data provided.

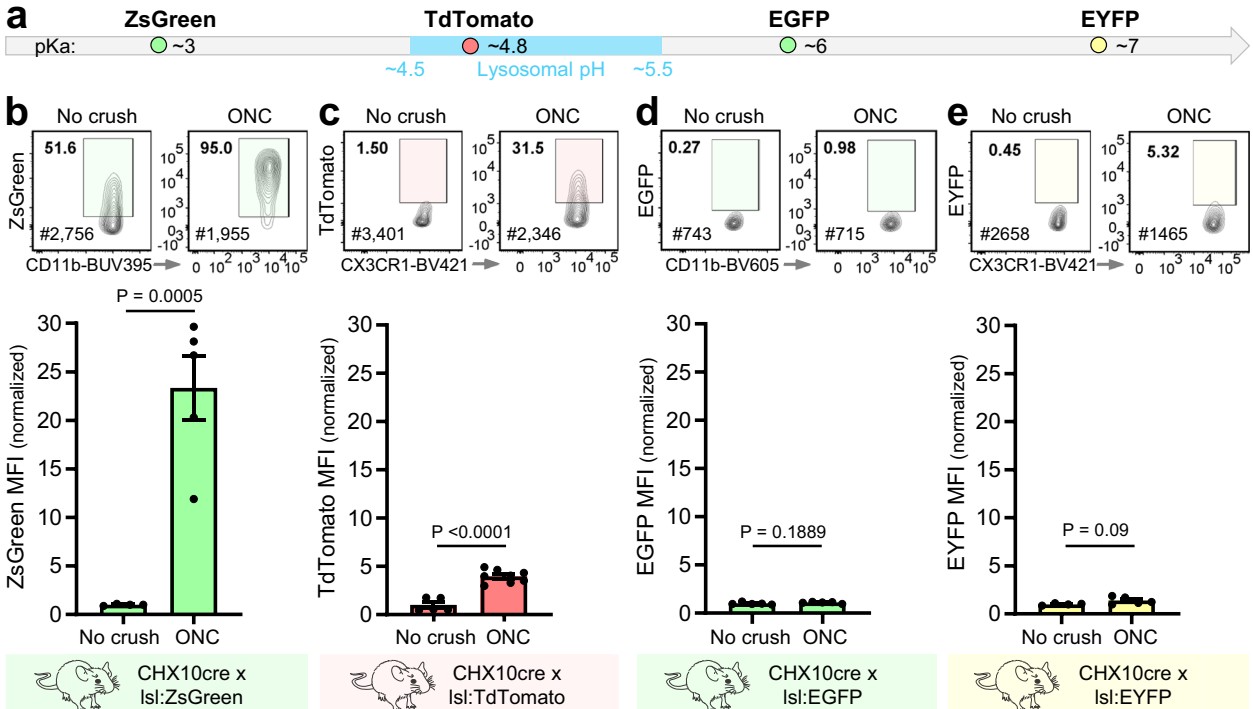

**Fig. 2 | In vivo engulfment of different fluorescent proteins following optic nerve crush. a** pKa values for fluorescent proteins (ZsGreen, TdTomato, EGFP, and EYFP) used to label retinal ganglion cells by crossing CHX-10-cre mice with lox-stop-lox (lsl) reporter mice. **b** MFI of ZsGreen contained in microglia from CHX10cre⁺lsl:ZsGreen mice. *n* = 5 (2 females and 3 males) for ONC and *n* = 4 (4 males) for No crush. **c** MFI of TdTomato contained in microglia from CHX10cre⁺lsl:TdTomato mice. *n* = 8 for ONC (2 females and 6 males) and *n* = 5 (5 females) for No crush. **d** MFI of EGFP contained in microglia from CHX10cre⁺lsl:EGFP mice. *n* = 5 for ONC (4 females and 1 male) and *n* = 5 for No crush (4 females and 1 male). **e** MFI of engulfed EYFP contained in microglia from

CHX10cre⁺lsl:EYFP mice. *n* = 5 (5 males) for ONC and *n* = 4 (2 females and 2 males) for No crush. **b–e** Microglia were isolated from LGNs/SCs by Dounce homogenization at 4 °C and enriched by Percoll centrifugation 3 days after bilateral ONC. Microglia are gated on live, single cells, CD45⁺, CD11b⁺, CX3CR1⁺, Ly-6C⁻ or CD206⁻. MFI values are normalized to the mean MFI of the No crush controls. The gates for positive microglia on the flow cytometric plots (5% contour plots) are based on the fluorescent signal from WT cortical microglia (<1% cells in positive gate) for each experiment. Error bars indicate standard error of the mean. Statistical analysis: unpaired two-tailed *t* test. Source data provided.

lysosomal acidification and should abate degradation of material engulfed in vivo (Fig. 4b). To assess debris contamination during the dissociation process, we mixed brain tissue from WT mice with brain tissue from SYN1-KO:ubiquitin-GFP (SYN1-KO:Ubi-GFP) mice before the dissociation step of FEAST. In this design, GFP⁻ microglia and BAMs are derived from WT brains containing SYN1, and therefore will contain SYN1 acquired in vivo and potentially SYN1 that was aberrantly acquired ex vivo. GFP⁺ microglia and BAMs (referred to as "sniffer cells") are derived from SYN1-KO brains and thus will only contain SYN1 material if they have acquired it aberrantly ex vivo from the WT brain they were mixed with during tissue dissociation or subsequent steps of FEAST (Fig. 4a, c).

We tested four dissociation conditions (Dounce homogenization, cold enzymatic digestion, Col-IV enzymatic digestion, and Col-IV enzymatic digestion + inhibitors) and quantified the amount of SYN1 protein in both endogenous WT microglia and BAMs, and in SYN1-deficient microglia and BAM "sniffer cells". This allowed us to identify distinct caveats of each dissociation approach.

As expected, we did not see SYN1 signal in "sniffer cells" that were dissociated in the absence of WT tissue ("KO" condition). However, we observed substantial ex vivo debris contamination of "sniffer cells" when KO and WT tissue were mixed together ("Mix" condition) and dissociated by Dounce homogenization or cold enzymatic digestion (Supplementary Data Fig. 2). These results prompted us to exclude these isolation methods for further development of FEAST to detect endogenous proteins.

We next compared SYN1 signals in cells that were isolated by Col-IV enzymatic digestion in the presence or absence of engulfment

inhibitors (Fig. 4a–e). Both WT microglia and BAMs digested in the absence of inhibitors showed substantially elevated SYN1 levels compared to cells digested in the presence of inhibitors. While microglial "sniffer cells" did not show significant debris contamination in either condition (Fig. 4d), BAM "sniffer cells" showed elevated false-positive signals independent of inhibitor use (Fig. 4e). These data suggest that both microglia and BAMs can aberrantly engulf synaptic material ex vivo and that BAMs acquire material by mechanisms that are partially insensitive to our inhibitor cocktail.

To pinpoint the step(s) of the isolation protocol that generate false-positive signals in BAMs, we designed an experiment in which we introduced the "sniffer cells" at different stages of isolation, labeling, and acquisition (Supplementary Fig 3). Brain tissue from WT mice and SYN1-KO:Ubi-GFP mice were either combined: prior to digestion (Mix 1); prior to fixation (Mix 2); or just before loading the samples on the cytometer for analysis (Mix 3). As observed previously, microglia "sniffer cells" showed minimal false-positive signals when mixed with WT cells at any step in the protocol. In contrast, BAM "sniffer cells" showed a substantial false-positive signal when mixed with WT cells before tissue digestion but not at subsequent steps in the protocol. We thus sought a more innovative approach that would allow us to prevent BAM (and microglia) acquisition of material at these early steps in the protocol.

## Harvesting cells from perfusion-fixed mice to overcome false-positive signal

To combat the issues of false-positive signals and to provide a robust platform for interrogating in vivo engulfment, we altered our strategy

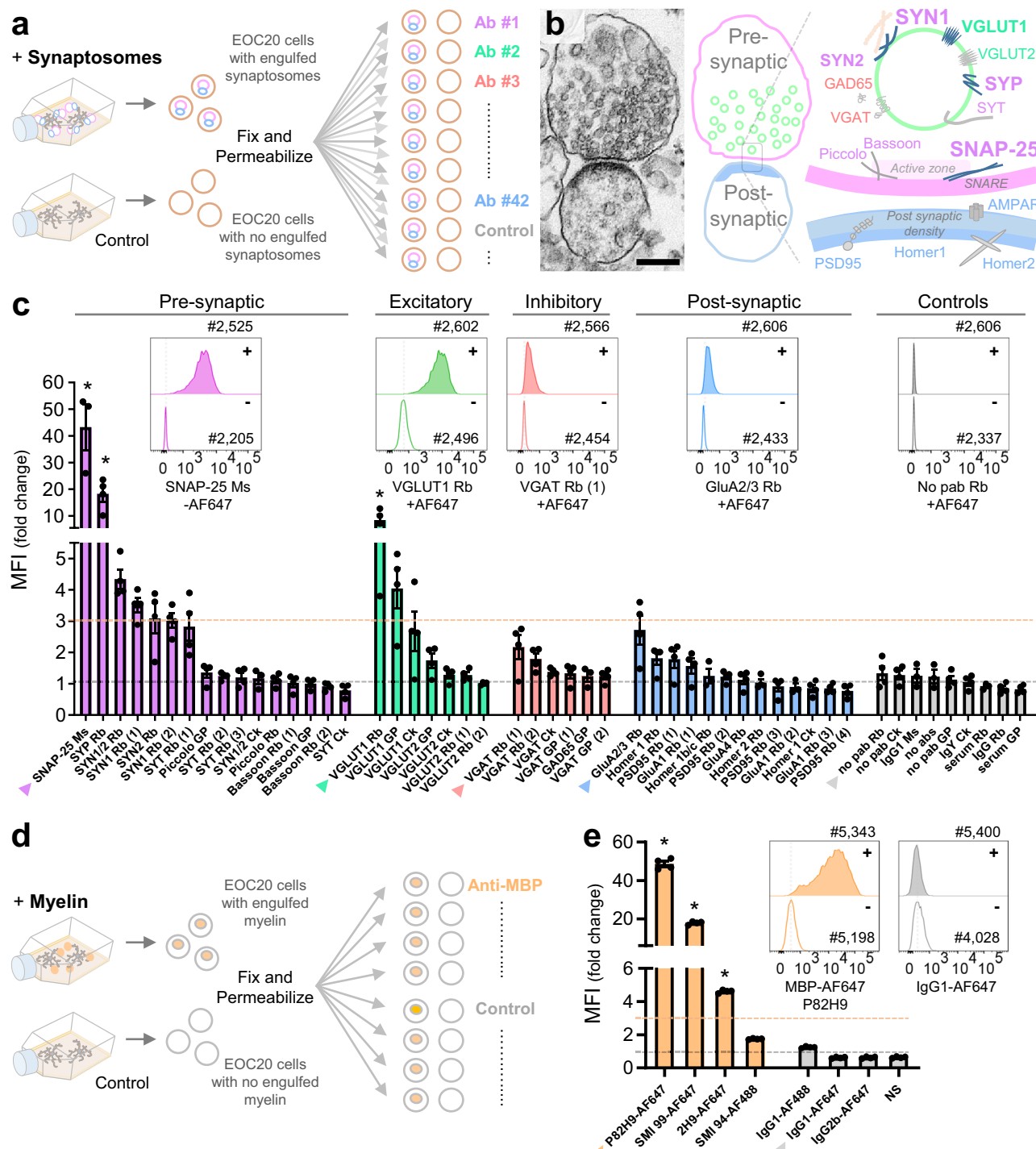

**Fig. 3 | Engulfment screen for antibodies against specific synaptic proteins and myelin basic protein. a** Experimental schematic depicting the screen for antibodies against synaptic proteins. Flasks with EOC20 cells were incubated with synaptosomes (+) or media (-) for 60 min. Cells were then harvested, surface labeled, fixed, permeabilized, and stained intracellularly with one of 42 different antibodies against synaptic proteins or appropriate isotype controls before being analyzed by flow cytometry. **b** Scanning electron microscopy image of a synaptosome containing pre- and post-synaptic elements used in the engulfment screen (left). Representative image of tens of synaptosomes examined. Scale bar represents 200 nm. Diagram of the synaptic proteins targeted in the engulfment screen (right). Proteins for which the engulfed MFI ratio was above three-fold are highlighted in bold. **c** MFI (fold change) for each antibody tested in the engulfment screen. The MFI for cells incubated with synaptosomes was divided by the MFI of untreated cells. The orange dotted line indicates an arbitrary threshold of three-fold change that was used to select candidate antibodies for further validation.

Histograms depict the staining of engulfed material corresponding to antibodies with the highest fold change in each category (also indicated with an arrowhead on the bar graph). $n = 4$ batches of cultured cells from four independent experiments, except for SNAP-25 Ms, SYN1/2 Ck, GAD65 GP, Homer 1b/c Rb, GluA1 Rb (2), and IgG1 MS for which $n = 3$. **d** Experimental schematic depicting a screen for antibodies against myelin basic protein (MBP). Flasks of EOC20 cells were incubated with myelin debris (+) or media (-) for 120 min and then processed for intracellular staining against MBP. **e** MFI (fold change) for each of the four different antibody clones against MBP and their respective isotype controls. Histograms for clone P82H9 and its IgG1 isotype control are displayed. $n = 4$ batches of cultured cells from four independent experiments. **c, e** Error bars indicate the standard error of the mean. Statistical analysis: One-way ANOVA, Bonferroni's multiple comparisons test comparing each antibody with its corresponding control. Non-significant $p$ values ($P \geq 0.05$) are not shown while * indicates $P$ values < 0.0001. Source data provided.

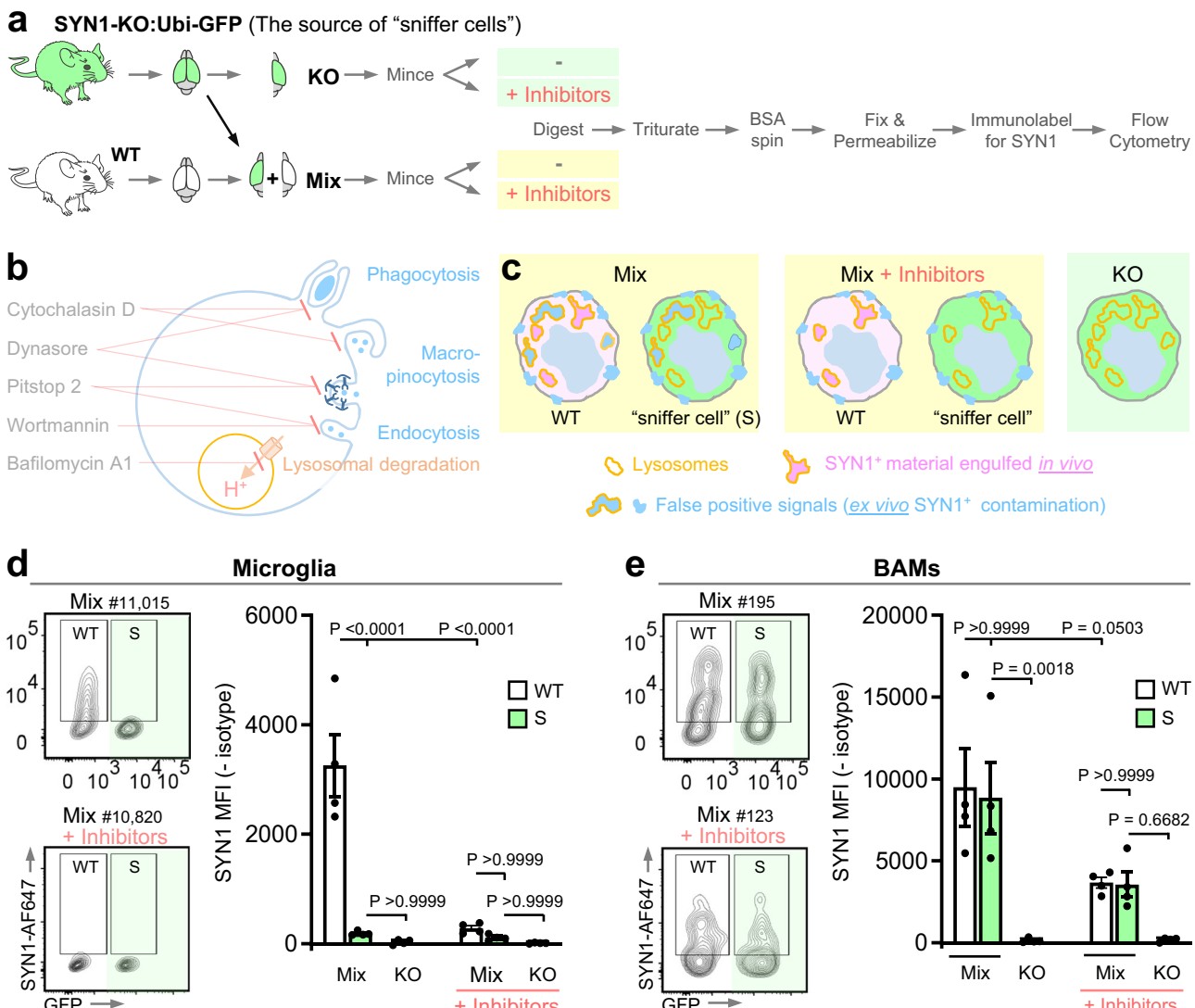

**Fig. 4 | Assessment of false-positive signal in microglia and border macrophages using "sniffer cells". a** Experimental schematic for assessment and abrogation of false-positive signal using "sniffer cells" and inhibitors of engulfment and lysosomal degradation. "Sniffer cells" for false SYN1 signal were introduced by mixing brain tissue from SYN1-KO mice (crossed with Ubi-GFP mice for identification) with brain tissue from WT mice. The samples were incubated with or without the cocktail of inhibitors and enzymatically digested with collagenase IV at 37 °C. Samples were then processed for flow cytometric analysis of SYN1. **b** Overview of the pharmacological inhibitor cocktail targeting engulfment and lysosomal degradation indicating the respective cellular processes targeted by each inhibitor. **c** SYN1-KO "sniffer cells" (shown in green) were introduced to reveal false-positive signals due to ex vivo SYN1+ contamination. The difference between WT cells and "sniffer cells" in the mixed samples without inhibitors is that the SYN1 signal in WT cells can be due to a combination of SYN1+ material engulfed in vivo (true signal, shown in pink) and ex vivo SYN1+ contamination (false-positive signal, shown in

blue). In contrast, any SYN1 signal in the "sniffer cells" can only be due to ex vivo SYN1+ contamination and represents a false-positive signal. Treatment with the inhibitor cocktail is designed to reduce/prevent false-positive signals due to ex vivo engulfment, while SYN1 signal from in vivo engulfment (true signal) should be unaffected. SYN1 signal was assessed in both microglia (**d**) and BAMs (**e**). The flow cytometry plots show the SYN1 signal for WT and "sniffer cells" (S) in the mixed samples (separated on the x-axis based on GFP expression). The gates for SYN1+ cells are based on the fluorescence of their respective isotype controls (<1% cells in positive gate). Cells were gated on live, single cells, CD45+, CD11b+, CD64+, and GR1- and microglia were further gated on CX3CR1$^{high}$ and P2Y12$^{high}$ while BAMS were gated on CD206$^{high}$ and CD38$^{high}$. n = 4 mice, all males, per condition. The mean fluorescent intensity (MFI) for SYN1 with isotype controls subtracted is depicted for all microglia (**d**) and BAMs (**e**) independent of the gating shown on the plots. Error bars indicate standard error of the mean. Statistical analysis: One-way ANOVA, Bonferroni's multiple comparisons test. Source data provided.

and aimed to harvest microglia and BAMs from perfusion-fixed mouse brains. Fixing the cells in vivo eliminates concerns about ex vivo engulfment and allows for the removal of contaminating debris with an acid wash of the isolated cells before labeling them for engulfed material (Fig. 5a). Like cells treated with our inhibitor cocktail, microglia and BAMs harvested from perfusion-fixed tissue showed a substantial decrease in engulfed SYN1 signal compared to cells isolated from fresh tissue by Col-IV digestion. The addition of an acid wash directly followed by a blocking step before permeabilization and staining for synaptic proteins further reduced the SYN1 signal in BAMs,

achieving 90% reduction in SYN1 signal compared to BAMs harvested from live tissue (Fig. 5b, c). The yield of microglia and BAMs was lower in the fix + acid wash group than in the live group (Supplementary Fig 4). However, unlike isolation by Dounce homogenization, the fix + acid wash protocol affected microglia and BAMs similarly, allowing us to analyze both cell types simultaneously with this method.

To validate our methodology and the engulfment of endogenous proteins in a robust in vivo paradigm, we returned to the ONC model. We isolated microglia and BAMs from perfusion-fixed mice 3 days after bilateral ONC and labeled them for SNAP-25 (Fig. 6a). As observed with

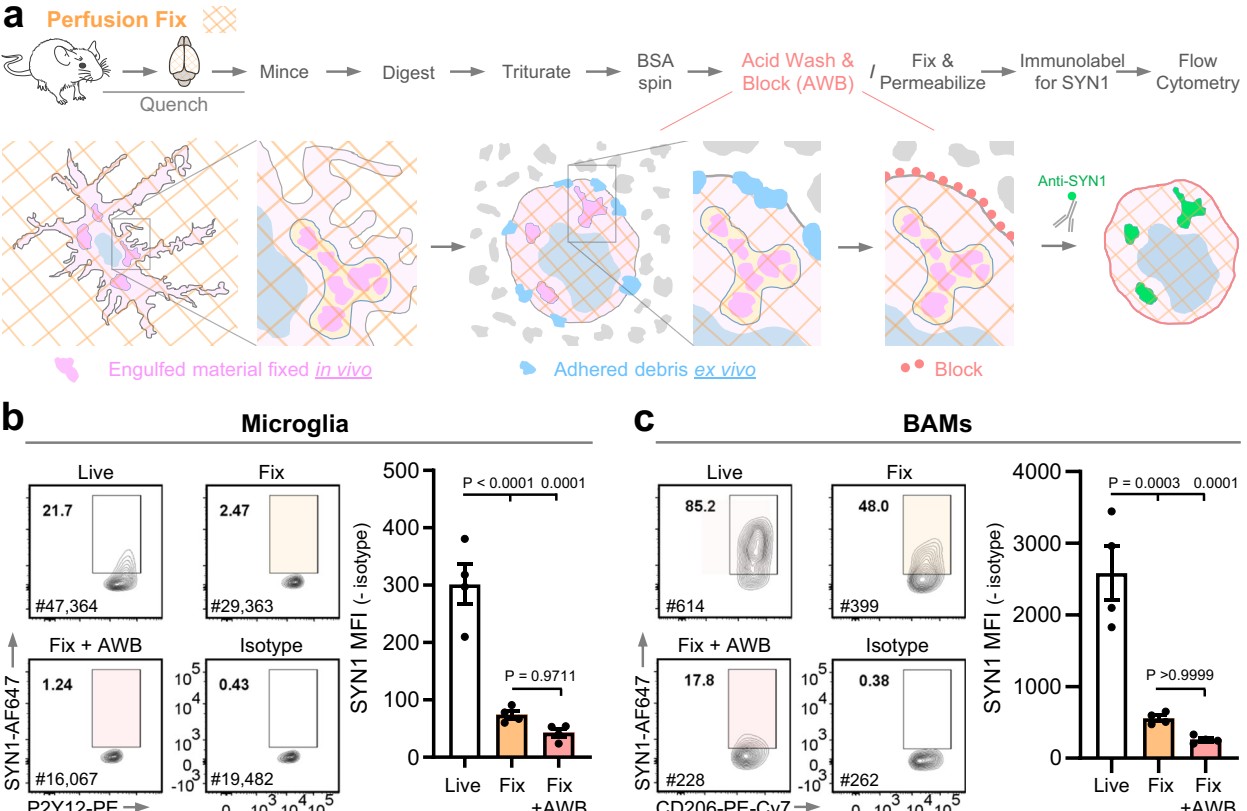

**Fig. 5 | Harvesting cells from perfusion-fixed tissue to reduce false-positive signal. a** Experimental schematic. Cells are harvested from perfusion-fixed tissue (fixed/cross-linked tissue is illustrated with an orange diagonal grid pattern). Residual paraformaldehyde is quenched prior to enzymatic digestion with Collagenase IV for 2 h at 37 °C. SYN1+ material engulfed in vivo prior to perfusion-fixation will be cross-linked inside the lysosomes (shown in light pink), while SYN1+ debris adhering to cells ex vivo (shown in blue) will detach after acid wash. Re-adherence of the debris is prevented with a blocking step (illustrated in red) prior to fluorescent immuno-labeling of SYN1 and analysis by flow cytometry. SYN1 signal was assessed in both microglia (**b**) and BAMs (**c**). The flow cytometry plots are displayed for SYN1 signal in cells harvested from live tissue, from fixed tissue with and without the acid wash and blocking step (AWB), and for the isotype control. The gates for SYN1+ cells are based on the fluorescence of their respective isotype controls (<1% cells in positive gate). Cells were gated on DAPI (to identify nucleated cells), single cells, CD45+, CD68+, GR- and microglia were further gated on CX3CR1high and P2Y12high while BAMS were gated on CD206high and CD38high. The MFI for SYN1 with the isotype control subtracted is depicted for all microglia (**b**) and BAMs (**c**) independent of the gates shown on the plots. n = 4 mice, all females, per condition. Error bars represent standard error of the mean. Statistical analysis: One-way ANOVA, Bonferroni's multiple comparisons test. Source data provided.

live microglia, ONC evoked a dramatic increase in AF488 signal in microglia compared to no crush (Fig. 6b). ONC also leads to an increase in engulfed SNAP-25 compared to no crush (Fig. 6c), demonstrating that we can successfully detect engulfment of endogenous synaptic proteins in fixed microglia. ONC did not evoke a significant increase in the MFI of AF488 or SNAP-25 in BAMs (Fig. 6d, e).

**Expanding FEAST to interrogate engulfment of myelin in two models of demyelination**

Our objective of developing FEAST was to establish a versatile methodology for interrogating microglial engulfment of synaptic material, and to concurrently establish a framework for examining engulfment of specific proteins across different tissues and paradigms at single cell resolution. To validate FEAST for detection of in vivo myelin engulfment, we isolated microglia and BAMs from perfusion-fixed mice bilaterally injected with lysolecithin (LPC) into the corpus callosum (CC) to trigger focal demyelination[22] (Fig. 7a–c), or from mice treated with cuprizone (CPZ) through dietary consumption to evoke global demyelination in the cortex (Cx) and the CC[23,24] (Fig. 7d–f). In LPC-injected mice, microglia from the CC showed increased engulfment of MBP relative to controls, whereas microglia from the Cx showed similar levels of MBP to controls. In mice treated with CPZ, microglia showed increased engulfment in both the CC and Cx compared to controls. In both the LPC-injected mice and in the CPZ-treated mice,

MBP+ microglia were associated with elevated CD68 expression. BAM engulfment of MBP did not change in the LPC or CPZ models (Supplementary Fig 5). These data reinforce that FEAST can sensitively detect cell type-specific engulfment of myelin selectively in the demyelinated brain regions across different paradigms.

## Discussion

We have developed and optimized FEAST, a straightforward and robust method to interrogate in vivo engulfment of synaptic and myelin material by flow cytometry. Critical features of FEAST include flexibility to detect engulfed material from labeled or unlabeled neurons or oligodendrocytes, integration of critical controls for artifactual engulfment, and scalability to quantify engulfment by hundreds to thousands of macrophages from numerous mice in a single experiment. We believe that FEAST provides distinct but complementary information about engulfment compared to field-standard imaging-based approaches, which facilitate examination of engulfment with higher anatomical specificity but at a much smaller scale.

Conceptually, the framework of FEAST can be applied to interrogate engulfment across nervous systems, including the peripheral and the enteric nervous systems and in different species, including human.

A requirement for interrogating engulfment in live cells is that neuronal material must be pre-labeled in vivo. This can either be

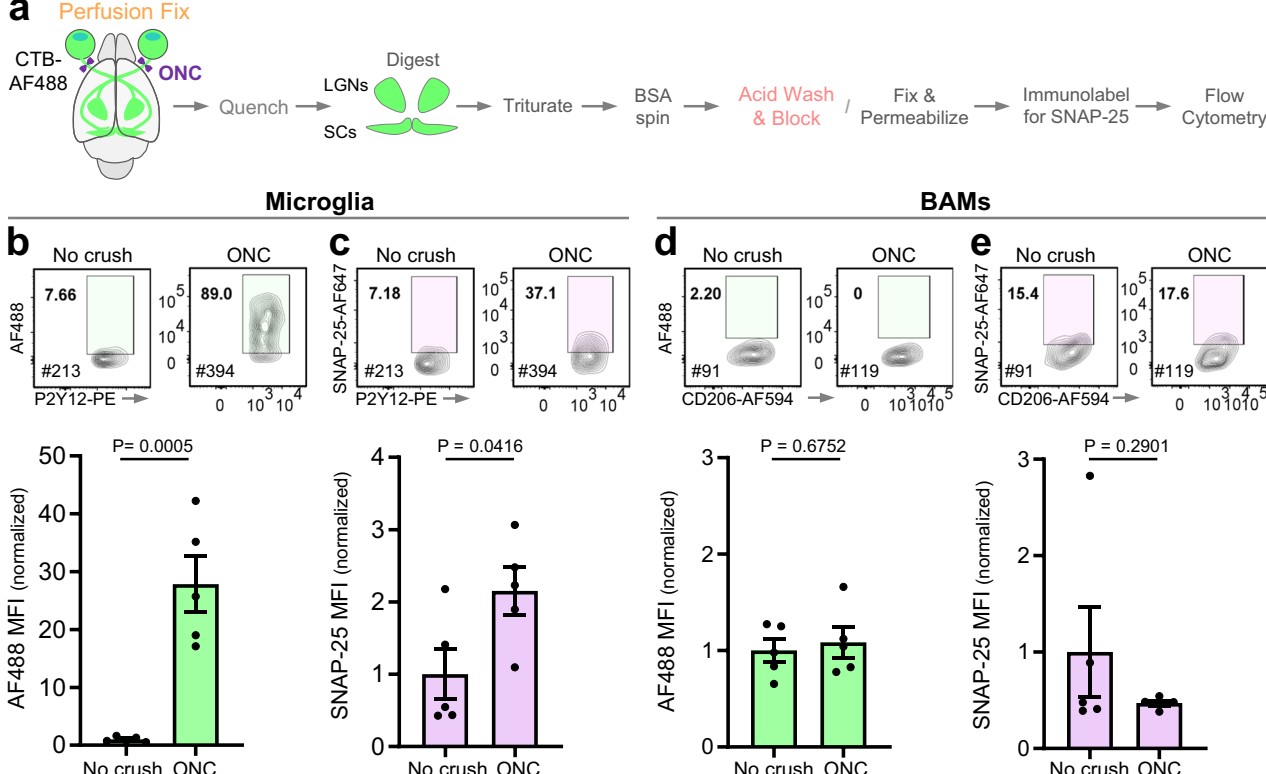

**Fig. 6 | Engulfment of endogenous synaptic proteins in the optic nerve crush paradigm by cells harvested from perfusion-fixed tissue. a** Experimental schematic depicting isolation of cells from perfusion-fixed tissue and assessment of engulfment following optic nerve crush (ONC). Prior to bilateral ONC, retinal ganglion cells in both eyes were labeled by intravitreal injection of cholera toxin subunit-B conjugated with AF488 (CTB-AF488) to facilitate the micro-dissection of the lateral geniculate nuclei and superior colliculi (LGNs/SCs). AF488 and SNAP-25 signals were compared between no crush and ONC for both microglia (**b**–**c**) and BAMs (**d**–**e**). Flow cytometry plots display AF488 and SNAP-25 signals in microglia and BAMs. The gates for AF488+ cells were based on the fluorescent signal from WT cortical cells while the gates for SNAP-25+ cells were based on the fluorescence of their respective isotype controls (<1% cells in positive gates). Cells were gated on DAPI (to identify nucleated cells), single cells, CD45+, CD68+, and GR1-. Microglia were further gated on CX3CR1high and P2Y12high, while BAMS were gated on CD206high and CD38high. The MFI for AF488 and SNAP-25 was normalized to the mean MFI of no crush controls and depicted for all microglia (**b**–**c**) and BAMs (**d**–**e**) independent of the gates shown on the plots. n = 5 mice (3 females and 2 males) per condition. Error bars depict standard error of the mean. Statistical analysis: unpaired two-tailed t test. Source data provided.

achieved by injection of a fluorescent tracer like CTB conjugated with a fluorescent dye or by viral or transgenic expression of FPs. Numerous transgenic mice and viruses allow the expression of a range of FPs, allowing circuit-specific labeling and the opportunity to address questions about the engulfment of those neuronal connections selectively. One obstacle to this approach is determining which FP to express. FPs sensitive to pH and/or lysosomal degradation are unsuitable for this purpose, as their signal becomes quenched upon engulfment. pH-stable FPs may be more suitable for engulfment analysis depending on their resistance to lysosomal proteases. These FPs include proteins such as RFP, mCherry, and ZsGreen. However, there are potential caveats in expressing stable FPs in vivo and using them to analyze engulfment. Accumulating stable FPs may be toxic to the neurons expressing them and consequently kill or alter the physiology of these neurons[25,26] leading to interpretations that might not accurately reflect engulfment of endogenous synaptic proteins. Additionally, analysis of engulfment of pH-stable FPs may obscure temporal aspects of this biology, as these proteins can survive for many hours or days in the endo-lysosomal compartment[27]. The use of FPs that fall within the range of endo-lysosomal pH, such as TdTomato, may allow a more accurate assessment of the in vivo biology of engulfment although with lower sensitivity.

To optimize a FEAST approach that could be applied in unmanipulated mice, we designed an in vitro screen with the goal of identifying a suite of highly specific antibodies that detect pre- or post-

synaptic proteins that have been engulfed by microglia. We were able to identify eight antibodies against five different pre-synaptic proteins that produced a reliable signal in the screen. The weak signal observed from antibodies against post-synaptic targets may result from excessive cross-linking and therefore reduced epitope availability of proteins in the postsynaptic density following fixation. Post-synaptic targets with epitopes outside of the post-synaptic density could be priorities in future screens, although we did test four different antibodies against homer proteins and four antibodies against different AMPA (α-amino-3-hydroxy-5-methyl-4-isoxazolepropionic acid) receptor subunits with limited success. Alternatively, different fixation protocols might be required. We also attempted to identify suitable antibodies for investigating engulfment of inhibitory synapses, screening five different antibodies against VGAT (vesicular γ-aminobutyric acid transporter). VGAT has previously been shown to be present in synaptosome preparations[17,28] but we were unable to detect engulfed VGAT in this assay. A flow cytometry panel that allows investigators to ask questions about engulfment of pre- versus post- and inhibitory versus excitatory synapses would be a valuable expansion of FEAST.

We also employed several different approaches to validate that the signal quantified by FEAST represents an accurate snapshot of the amount of synaptic material that each microglia or BAM had engulfed in vivo. We reasoned that there would be two potential sources of contamination: 1) synaptic material that was actively engulfed ex vivo

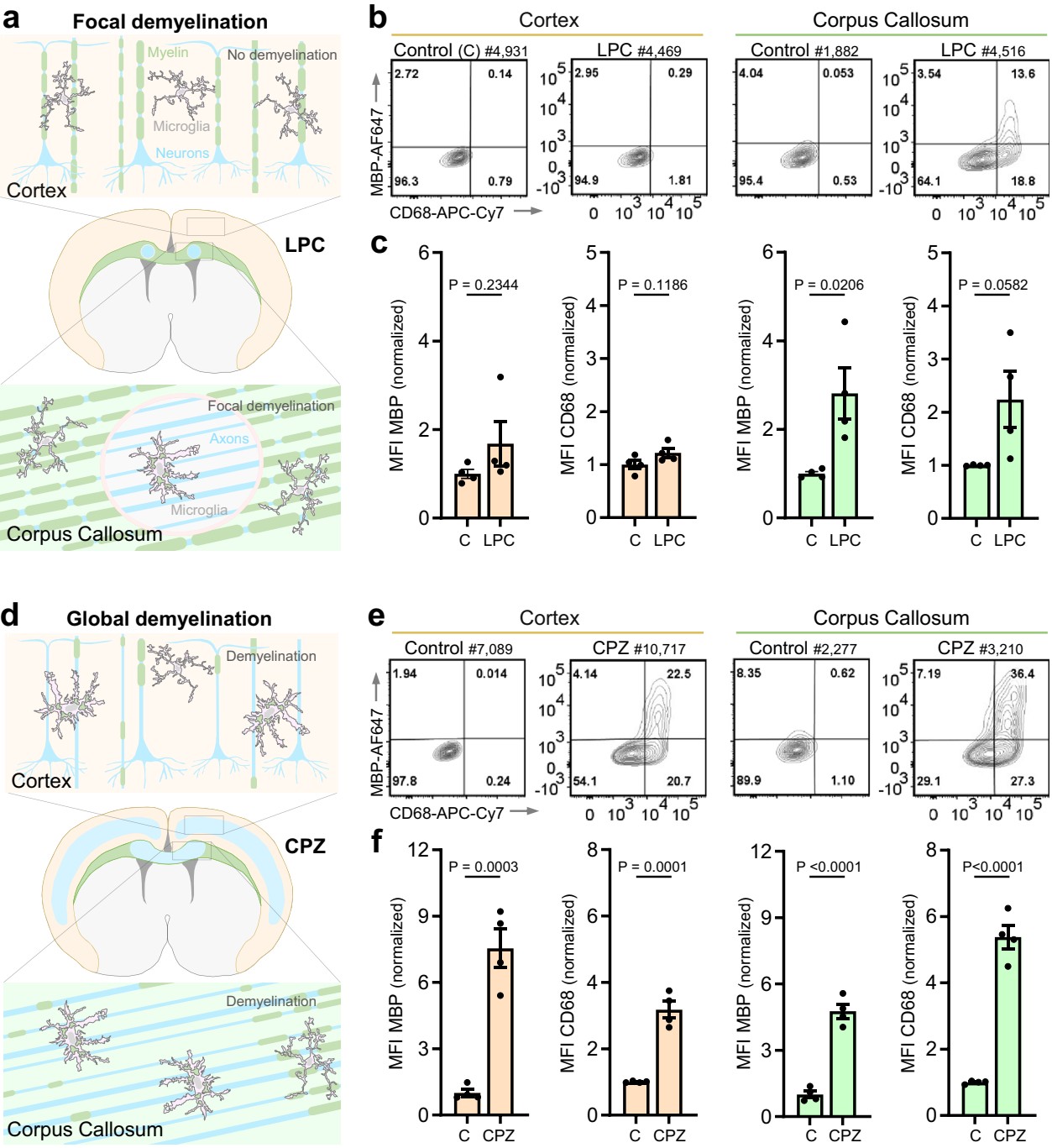

**Fig. 7 | Myelin engulfment following focal or global demyelination by cells harvested from perfusion-fixed tissue. a** Experimental schematic of the focal demyelination paradigm. Bilateral injection of lysolecithin (LPC) into localized areas of the corpus callosum (shown in green) results in focal demyelination (indicated in light blue) with no demyelination in the cortex (shown in light orange). **b** Microglia were harvested from perfusion-fixed tissue 7 days after injection of LPC. Flow cytometry plots display MBP and CD68 signal in microglia for both the cortex and the corpus callosum in control (non-injected) and LPC-injected mice. **c** MFI of MBP and CD68 signals in all microglia from cortex or corpus callosum (normalized to control). **d** Experimental schematic of the global demyelination paradigm. Treatment with cuprizone (CPZ, in diet) results in demyelination of both the cortex

(shown in light orange with demyelination indicated in light blue) and the corpus callosum (shown in green with demyelination indicated in light blue). **e** Microglia were harvested from perfusion-fixed tissue after 3 weeks of CPZ treatment. Flow cytometry plots display MBP and CD68 signal in all microglia from cortex or corpus callosum in control (fed control diet) and CPZ-treated mice. **f** MFI of MBP and CD68 signal in all microglia from cortex or corpus callosum (normalized to control). Microglia were gated on DAPI (to identify nucleated cells), single cells, CD45$^+$, CD68$^+$, GR1$^-$, CX3CR1$^{high}$, and P2Y12$^{high}$. $n$ = 4 mice per condition: (**c**) 4 males and (**f**) 2 females and 2 males. Error bars indicate standard error of the mean. Statistical analysis: unpaired two-tailed t-test. Source data provided.

during the tissue preparation process, and 2) material that passively adhered to the outside of the cells or to intracellular compartments after fixation and permeabilization. Our unilateral crush experiments showed that inputs from crushed axons were engulfed selectively with minimal contamination from uncrushed axonal material when microglia were isolated from live tissue and kept alive during subsequent isolation and analysis. Additionally, we did not observe a signal from pH-sensitive FPs (e.g., GFP, YFP) in microglia after ONC and analysis of live microglia. These data indicate that the cold Dounce homogenization protocol used for examination of live microglia is sufficient to prevent ex vivo engulfment of ambient debris and that our Percoll purification protocol is sufficient to prevent excessive fluorescent debris from adhering to the surface of live microglia in this paradigm.

Our FEAST protocol for endogenous synaptic and myelin proteins required two major modifications—enzymatic digestion and fixation/permeabilization - compared to the protocol we used for the evaluation of live cells. Since both modifications may alter ex vivo acquisition of debris, we implemented "sniffer cells" (cells from SYN1-KO mice that cannot have engulfed SYN1 in vivo) as an experimental method to assess false-positive signals during optimization of these aspects of FEAST. We also introduced a cocktail of engulfment and lysosomal inhibitors to reduce the possibility for active acquisition or degradation of material ex vivo during the FEAST protocol. The "sniffer cell" approach allowed us to determine that microglia acquire substantial false-positive SYN1 signal when dissociated by Dounce homogenization and then fixed, permeabilized, and stained intracellularly for engulfed material. This data contrasts to what we observed for microglia isolated by Dounce homogenization and then kept alive to analyze engulfment of pre-labeled neuronal material, suggesting that fixed/permeabilized cells may be more vulnerable to adherence of false-positive debris in certain contexts.

Additionally, by combining the "sniffer cell" and inhibitor approaches, we were able to demonstrate that both microglia and BAMs can rapidly acquire synaptic material ex vivo (likely during the enzymatic digest at 37 °C) and that this acquisition can be blocked in microglia and mitigated only partially in BAMs by addition of engulfment and lysosomal inhibitors. The "sniffer cell" approach also allowed us to identify an additional major caveat with the application of FEAST to examine engulfment by BAMs. BAM "sniffer cells" showed equivalent, high levels of synaptic material compared to endogenous WT cells in both the control and inhibitor conditions, indicating that BAMs acquire material by additional mechanisms that we could not address with this approach. This is likely a combination of inhibition-resistant engulfment and adherence of debris to an array of endocytic receptors that are enriched in BAMs compared to microglia (e.g., lectin receptors and glycan receptors).

To prevent any possibility of ex vivo engulfment, and to allow for downstream removal of adhered debris from the surface microglia and BAMs, we modified the FEAST protocol to allow isolation of microglia and BAMs from perfusion-fixed tissue and added a subsequent acid wash and blocking step. While this approach decreased the yield of both microglia and BAMs, it allowed us to remove a substantial amount of ex vivo contamination from both microglia and BAMs. We were able to use this protocol to sensitively and specifically detect differences in microglial engulfment of synaptic protein in the ONC paradigm and of myelin protein in two models of demyelination. Interestingly, engulfment of material by BAMs was not altered in either of these paradigms, supporting a biological division of labor between microglia embedded in the brain parenchyma and BAMs positioned at the borders of the brain.

Conceptually, FEAST is not restricted to engulfment of synaptic or myelin material and can be expanded to quantify the acquisition of any type of material by any cell type. Some future applications of interest for the field of neuro-immunology might include engulfment of amyloid-beta by brain resident versus infiltrating immune cells in

models of Alzheimer's disease with or without a pharmaceutical intervention, or engulfment of myelin by macrophages versus dendritic cells during experimental autoimmune encephalomyelitis. It will also be informative to tease apart the physiological roles of microglia versus BAM engulfment using FEAST, as these cell types likely survey different pools of CNS substrates. Indeed, perivascular BAMs rapidly engulf fluorescently labeled dextran or ovalbumin injected into the CSF, and meningeal BAMs can engulf dyes injected into both the CSF and the blood[29]. While these data are compelling, it is unknown whether BAMs engulf endogenous substrates at steady-state and whether engulfment by BAMs contributes to the maintenance of brain physiology in the healthy or diseased CNS. We have provided a template for optimization of FEAST and encourage other groups to apply a similar methodology and equally rigorous sets of controls when applying FEAST in a new biological system.

Beyond assessment of engulfment, FEAST might also be leveraged to elucidate the molecular mechanisms driving microglia-neuron interactions in different paradigms. For example, a future utility of FEAST is the potential of isolating microglia via fluorescence activated cell sorting (FACS) based on how much neuronal material they have engulfed in vivo. This approach is timely for the era of single-cell transcriptomics and other –omic analyses and opens the possibility of combining a functional readout with these techniques. Synapses could also be 'barcoded' by unique oligonucleotide sequences[30,31] in vivo and engulfing microglia could be sorted for single-cell transcriptomic analysis. This approach would allow an analysis of the correlation between the transcriptional profile of the microglia and the number or cellular origin of engulfed barcodes on a single cell basis. Alternatively, advancements allowing single-cell RNA sequencing of fixed cells could be followed by a combined FEAST/CITE-seq[32] approach with barcoded antibodies targeting engulfed endogenous proteins such as SNAP-25 and MBP. Future strategies for FEAST could also include translational incorporation of non-canonical amino acids into specific neuronal populations followed by click chemistry-mediated fluorescent labeling of those amino acids in microglia post-isolation[33–35]. A complete profile of all engulfed synaptic proteins could be determined by combining this approach with mass spectrometric analysis of engulfed peptides containing non-canonical amino acids, facilitating the exploration of more specific biological questions about microglial engulfment of synaptic terminals.

## Methods
### Mice
All housing and experimental procedures were approved and overseen by Boston Children's Hospital Institutional Animal Care and Use Committee following NIH guidelines for the humane treatment of animals. Animal license numbers: 17-09-3524R, 20-05-4159R, and 20-08-4236R. Mice were group housed in Optimice cages under a 12-h light/12-h dark cycle and maintained in the temperature range and environmental conditions recommended by AAALAC with food and water ad libitum. All mice used in this study were from in-house breeders except for the mice used for demyelination studies. For those experiments (LPC injection and CPZ treatment), adult C57BL/6J mice (JAX stock No: 000664) and adult C57BL/6NJ (JAX stock No: 005304) were purchased from JAX Laboratory and housed in our animal facility for 7 days before experiments were initiated. To investigate engulfment of fluorescent proteins CHX10-Cre (Tg(Chx10-EGFP/cre,-ALPP) 2Clc/J: JAX Stock No: 005105) mice were crossed with four different fluorescent reporter lines: lsl:ZsGreen (B6.Cg-Gt(ROSA)26Sortm6(-CAG-ZsGreen1)Hze/J: JAX Stock No: 007906), lsl:TdTomato (B6.Cg-Gt(ROSA)26Sortm9(CAG-tdTomato)Hze/J: JAX Stock No: 007909), lsl:EGFP (B6.129(Cg)-Gt(ROSA)26Sortm4(ACTB-tdTomato,-EGFP)Luo/ J: JAX Stock No: 007676), and lsl:EYFP (B6.129X1-Gt(ROSA)26Sort-m1(EYFP)Cos/J: JAX Stock No: 006148). SYN1-KO synaptosomes were generated from SYN1-KO mice (B6.129P2-Syn1tm1Pggd/Mmja), JAX

Stock No: 41436-JAX and these mice were also crossed with Ubi-GFP (C57BL/6-Tg(UBC-GFP)30Scha/J) JAX Stock No: 004353 to generate SYN1-KO:Ubi-GFP mice for the "sniffer cell" experiments.

Both male and female mice were used in this study.

## Intravitreal injections and optic nerve crush

Female and male mice (older than P60) were anesthetized with isoflurane and kept under constant isoflurane exposure throughout the procedure. After mice were no longer responsive to a toe pinch, the tip of a 30G needle (BD Precision Glide: Cat# 305111) was inserted through the sclera. The 30G needle was removed and a 33G blunt-end Hamilton needle (Hamilton; Cat# 7803052) attached to a Hamilton syringe (VWR; Cat# 60376172) was inserted and used to deliver 3 µl of 5 mg/ml CTB subunit into the vitreous fluid (~0.1 µl/sec). For bilateral ONC experiments, CTB-AF488 (Life Technologies; Cat# C22841) was injected into both eyes. For unilateral ONC experiments, CTB-AF488 was injected into the left eye of each mouse and CTB-AF594 (Life Technologies; Cat# C22842) was injected into the right eye.

Four days after intravitreal injection of CTB, mice were anesthetized with Ketamine (100 mg/kg)/Xylazine (10 mg/kg) and positioned in a stereotaxic apparatus (Kopf Instruments, Tujunga, CA). The optic nerve was exposed and crushed ~1 mm behind the eyeball with angled jeweler's forceps (Dumont # 5) for 5 s.

## LPC injections and CPZ treatment

Focal demyelination was evoked in P90 mice by bilateral injections of LPC. Mice were anesthetized with Ketamine/Xylazine and positioned in a stereotaxic apparatus. Using a Nanoject III (Drummond, Cat# 490019-810), 500 nL of 1% LPC (EMD chemicals, Cat#: 440154) were injected at a flow rate of 1 nL/s at the following coordinates from bregma: 1.2 mm anterior to Bregma, ± 0.5 mm lateral, 1.4 mm deep, normalized to the surface of the skull. Non-operated littermates were used as controls.

Global demyelination was induced by providing P90, C57BL/6NJ mice 18% protein Rodent Diet containing 0.2% CPZ (Envigo Teklad Global, Cat# TD.140803) for 3 weeks on an ad libitum basis. Food was topped off three times per week and was fully changed once weekly. The food stock was kept vacuum-sealed to mitigate the effects of moisture on the CPZ efficiency. Littermates that were fed 18% protein rodent diet (Envigo Teklad Global, Cat# T.2018.15) without CPZ served as controls.

## Preparation of single-cell suspensions from live tissue

We isolated live microglia and BAMs using one of three approaches: Dounce homogenization; enzymatic digestion at 37 °C with Collagenase IV; or enzymatic digestion at 4 °C with a cold-active protease from *Bacillus licheniformis* (each described below). All solutions were pre-made and stored at 4 °C or on ice. All tubes, Dounce homogenizers, and plates containing tissue or cells were kept on ice, and centrifuges were pre-cooled to 4 °C to minimize cell death and artifactual ex vivo engulfment.

**Euthanasia.** For all approaches, mice were anesthetized by intraperitoneal injection with 40 µl per g body weight of Avertin (2,2,2-Tribromoethanl (Millipore Sigma: Cat# T48402) dissolved in 2-Methyl-2-butanol (Millipore Sigma; Cat# 240486) and diluted to 20 mg/ml in Hank's balanced salt solution (HBSS, Life Technologies; Cat# 14175-145). When no toe pinch reflex was observed, mice were transcardially perfused with 1 ml cold HBSS per gram body weight and the brains were removed and submerged in ice cold HBSS.

**Dounce homogenization (4 °C).** For the ONC experiments, the LGNs and SCs were microdissected, aided by their fluorescent signal, using a fluorescent dissection scope. The two LGNs and the two SCs from each mouse were collected and pooled together in 1 ml FACS buffer

(HBSS + 0.5% BSA (bovine serum albumin), Millipore Sigma; Cat# A2153) and 2 mM EDTA (Fisher Scientific; Cat# 15-575-020) in a glass tissue grinder (DWK Life Sciences, Kimble; Cat# 885300-0002) and Dounce-homogenized by 20–25 gentle strokes with pestle "A" and 25–30 strokes with pestle "B". For the cerebrum experiments presented in Figs. 4–5 and Supplementary Figs. 2–4, the whole brain was isolated, the olfactory bulbs, brain stem, and cerebellum were removed, and the cerebrum was minced with razor blades (Fisher Scientific; Cat# 12–640) and then transferred to a tissue grinder for Dounce homogenization. The homogenates were filtered into 15 ml tubes through a 70 µm cell strainer (Fisher Scientific; Cat# 22-363–548) and the volume was adjusted to ~5.5 ml with FACS buffer. 4.5 ml of 90% isotonic Percoll density gradient media (Millipore Sigma; Cat# GE17-0891-01) diluted with 10× HBSS (Thermo Fisher Scientific; Cat# 14185052) was added, and the tubes were gently inverted to make a total of 10 ml mixture of the brain homogenate in 40% Percoll solution. Microglia were enriched by centrifugation for 1 h at $500 \times g$ at 4 °C. The myelin layer on top and most of the supernatant were removed using a vacuum pump, and the last remainder was removed with a P1000 pipette. The pellet was resuspended in 1 ml FACS buffer and transferred to a 96-deep well plate (Protein LoBind, Eppendorf; Cat# EP0030504305).

**Enzymatic digestion with Collagenase IV (37 °C) or *Bacillus licheniformis* protease (4 °C).** For the ONC experiments, LGNs and SCs were transferred to a 2 ml tube containing 2 ml of RPMI-H ((Gibco RPMI 1640 Medium, no phenol red, Fisher Scientific; Cat# 11835-055) + 10 mM HEPES (Millipore Sigma; Cat# 83264)) and chopped with scissors (Roboz Surgical instrument; Cat# RS-5912). The cerebra were transferred to a Petri dish and minced with razor blades until they resembled a fine paste.

Tissues to be digested with Collagenase IV (Col-IV) at 37 °C (Figs. 4–5) were transferred to a 5 ml capped FACS tube and pre-incubated for 20 min in 4 ml RPMI-H + engulfment/lysosomal inhibitor cocktail (described in Engulfment/Lysosomal Inhibitor Cocktail section) while gently agitating at 4 °C. The samples were then centrifuged at $300 \times g$ for 3 min to pellet tissue before addition of digestion cocktails. The supernatant was discarded, and pellets were resuspended in 4 ml of pre-heated digestion mix (RPMI-H + Collagenase IV (800 U/mL, Worthington; Cat# LS004189) + DNAse-1 (250 U/mL, Worthington; Cat# LK003172)) plus the inhibitor cocktail and incubated for 45 min at 37 °C under constant, gentle agitation.

Tissues to be digested with *Bacillus licheniformis* protease at 4 °C (Supplementary Fig 2) were transferred into a 5 mL capped FACS tube (Fisher Scientific; Cat# 149591 A) and 4 ml of a cold digestion cocktail (RPMI-H + *Bacillus licheniformis* protease (10 mg/ml, Millipore Sigma; Cat# P5380) + *Serratia marcescens* nuclease (250 U/mL, Kerafast; Cat# EVC010)) was added. Samples were incubated for 45 min at 4 °C under constant, gentle agitation.

All samples were then centrifuged at $500 \times g$ for 5 min, the supernatant discarded, and 1 ml cold FACS buffer added to stop the digestion. Each sample was triturated with a P1000 pipette (10–20 times lightly followed by 20 times with more force). Homogenates were filtered into 15 ml tubes through a 70 µm filter and filled to 12 ml with FACS buffer. Samples were centrifuged at $500 \times g$ for 5 min, the supernatant decanted, the pellet resuspended in 1 ml FACS buffer by pipetting with a P1000, and the volume adjusted to 2 ml with FACS buffer. A density spin with BSA was applied to enrich for microglia after digestion. 5 ml of 25% BSA (25% w/v in HBSS) was added, the tubes were inverted to mix the homogenate and the BSA to a final BSA concentration of ~18%, and the samples were then centrifuged at $1200 \times g$ for 10 min. The myelin layer on top and supernatant was removed using a vacuum pump. The pellet was then resuspended in 2 ml FACS buffer and transferred to a 2 ml round bottom tube (Protein LoBind, Eppendorf Cat# 022431102) and centrifuged at $500 \times g$ for 8 min to wash off

residual BSA. Finally, the supernatants were decanted, the pellets were resuspended in 100 µl FACS buffer (for a total volume of ~200 ul), and the samples were transferred to a 96-well U-bottom polypropylene plate (Thermo Fisher Scientific Cat# 267334).

## Surface labeling of live cells in suspension

Cells were pelleted by centrifuging the 96-well plate at $500 \times g$ for 5 min and the supernatant decanted. Next, the cells were resuspended in 50 µl HBSS + 2 mM EDTA containing Fc block and viability dye. After 20 min, 50 µl of FACS buffer containing a cocktail of antibodies designed to distinguish microglia and BAMs (Supplementary Table 2) was added and samples were incubated for an additional 20 min. 100 µl of FACS buffer was added and the samples were centrifuged at $500 \times g$ for 5 min followed by a wash and spin in 200 µl FACS buffer. Finally, cells were resuspended in 200 µl of FACS buffer and filtered into a 5 ml tube (Polystyrene tubes with cell strainer cap, Corning Life Science; Cat# 352235) for flow cytometric acquisition. Alternatively, cells were resuspended in fixation buffer to allow subsequent intracellular staining for synaptic proteins.

## Fixation, permeabilization, and intracellular staining of cells in suspension

Following the surface labeling and a wash in FACS buffer, cells were fixed by the addition of 100 µl cold fixation buffer (eBioscience IC Fixation Buffer, Thermo Fisher Scientific; Cat# 00-8222-49) and incubation at room temperature (RT) for 20 min. From this point on, each step was performed at RT and with solutions at RT. 100 µl of FACS buffer was added and the cells were centrifuged at $500 \times g$ for 5 min. The supernatant was decanted, and the cells were washed by resuspending them in 200 µl FACS buffer followed by a spin at $500 \times g$ for 5 min. The fixed cells were either resuspended in 200 µl FACS buffer, and the plate covered with an adhesive sealing film (Bio-Rad; Cat# MSB1001) and stored at 4 °C for use within the next week, or immediately permeabilized for intracellular staining.

Cells were resuspended in 200 µl of eBioscience Permeabilization Buffer (Thermo Fisher Scientific; Cat# 00833356) diluted based on the manufacturer's instructions and centrifuged at $500 \times g$ for 5 min. Cells were subsequently blocked in 50 µl of permeabilization buffer with 10% goat serum (Millipore Sigma; Cat# G9023) for 20 min. 50 µl of permeabilization buffer containing either anti-SYN1 or anti-SNAP-25 (Supplementary Table 2) was then added directly to the samples. The solution was mixed by pipetting up and down and allowed to incubate for an additional 20 min. 100 µl permeabilization buffer was added and the cells were centrifuged at $500 \times g$ for 5 min. Cells were washed in an additional 200 µl permeabilization buffer and pelleted at $500 \times g$ for 5 min. If the synaptic antibody was primary conjugated, the pellets were then resuspended in 200 µl FACS buffer and filtered into a 5 ml tube for flow cytometric analysis. If the primary antibody was unconjugated, as in the case of the antibody screen, the pellets were resuspended in 100 µl of permeabilization buffer with 5% goat serum containing the appropriate AF647-conjugated secondary antibody (Supplementary Table 1) and incubated for 20 min. Cells were washed by addition of 100 µl permeabilization buffer and centrifugation a $500 \times g$ for 5 min, followed by a wash and spin in 200 µl permeabilization buffer. Finally, the cells were resuspended in 200 µl FACS buffer and filtered into a 5 ml tube for flow cytometric analysis.

## Harvesting and staining cells after perfusion-fixation

Mice were deeply anesthetized with Avertin and transcardially perfused with 10 ml of cold HBSS followed by 20 ml of BD ICC Fixation Buffer (Cat# BD 550010) at a flow rate of 7 ml/min. Note that Fixation Buffer (Biolegend Cat# 420801) can be used as a substitute for The BD ICC Fixation Buffer.

The brains were removed and submerged in a solution (at RT) containing 250 mM glycine (Sigma-Aldrich, Cat# G8898), 250 mM Tris (Millipore Sigma, Cat# GE17-1321-01), 5 mM HEPES, and 0.1% NaN$_3$ (Millipore Sigma, Cat# 71290) dissolved in HBSS, designed to quench residual paraformaldehyde (PFA). The brain regions of interest were then dissected; cerebrum (Fig. 5), LGN and SC (Fig. 6) Cx and CC (Fig. 7). The tissues were then minced with razor blades or chopped with scissors and enzymatically digested with Col-IV for 2 h at 37 °C following the steps described above. Inhibitors were omitted as the fixation eliminates the possibility of ex vivo engulfment. DNAse was omitted as we wanted to preserve DNA to identify nucleated cells as part of the flow cytometric assessment. The digested tissue was triturated and microglia and BAMs were enriched by a density spin with BSA at $1200 \times g$ for 10 min. The cells were transferred to a 96-well U-bottom plate and pelleted, then subjected to an acid wash by resuspension in 200 µl of a solution with 0.2 M acetic acid (Millipore Sigma, Cat# A6283) in 0.9% saline (Milipore Sigma, S8776) pH 2.75. After 15 min of incubation on ice, the cells were spun at $500 \times g$ for 5 min at 4 °C. The supernatant was decanted and the pellet was resuspended in 200 µl of a solution containing 20% Ovalbumin (Millipore Sigma, Cat# A5503), 20% mouse serum (Millipore Sigma, Cat# M5905), 5 mM HEPES, and 0.1% NaN$_3$ designed to block debris from re-adhering. After 15 min of incubation at RT, the cells were spun at $800 \times g$ for 5 min, the supernatant decanted, and the cells were fixed and permeabilized as described above. Note that the perfusion-fixation only offers a light fixation which is compatible with generating a single cell suspension, preventing ex vivo engulfment, and enduring the acid wash. The additional fixation step is required prior to permeabilization or for storing the cells.

The staining of cells was carried out by resuspending them in 50 µl of permeabilization buffer with 10% goat serum and Fc Block. After 20 min, 50 µl of a cocktail of antibodies and DAPI (4',6-Diamidino-2-Phenylindole, Biolegend; Cat# 422801) in permeabilization buffer was added (Supplementary Table 3). The solutions were mixed and allowed to incubate for an additional 20 min. After two additional washing steps in permeabilization buffer, the cells were resuspended in 100 µl FACS buffer and filtered into a 5 ml tube for flow cytometric analysis.

A schematic overview of the three main approaches that we have undertaken to interrogate in vivo engulfment is outlined in Additional Supplementary Information 1 followed by detailed step-by-step protocols (Additional Supplementary Information 2, 3, and 4).

## Flow cytometric acquisition

All samples except for those containing TdTomato were analyzed on a FACSAria SORP (special order research product) II flow cytometer with FACSDIVA version 8.0.1 software. Up to five different lasers and twelve bandpass filters (bp) (Supplementary Table 4) were used for the detection of each staining in the designed panel (Supplementary Tables 2, 3). Samples containing TdTomato were analyzed on a FACSAria II flow cytometer with a 561 nm laser. Fluorescent beads (Rainbow fluorescent particles, BD Bioscience; Cat# 556291) were run before each experiment to optimize the laser delays and again after each experiment to confirm that the lasers had been stable throughout the entire experiment. All settings including flow speed were kept consistent and each sample was diluted to allow acquisition of 800–1200 events per second with a FSC-A threshold of 5000 applied. All samples were run at 70 psi through a 70 µm nozzle (BD Bioscience; Cat# 643940). For equal sample acquisition across different groups, counting beads (CountBright, Thermo Fisher Scientific; Cat# C36950) were added prior to the flow cytometric assessment (~50,000 beads per sample). The gating strategies, including FMO (fluorescence minus one) controls are displayed in the Supplementary Figs. 6, 7. Representative gating strategies used for Fig. 1d, e and Fig. 2b–e are provided in Supplementary Fig 8.

## Engulfment / lysosomal inhibitor cocktail

A cocktail of inhibitors was designed to block ex vivo engulfment. Cytochalasin D (2 µM, Tocris; Cat# 1233), Wortmannin (2 µM, Tocris; Cat# 1232), Pitstop 2 (25 µM, Abcam; Cat# ab120687), and Dynasore (40 µM, Tocris; Cat# 2897) were applied to block phagocytosis, macropinocytosis, and endocytosis (clathrin-, caveolin-dependent and independent). Additionally, Bafilomycin A1 (40 nM, Tocris; Cat# 1334) was applied to inhibit the acidification of endosomes and lysosomes by targeting vacuolar-type H + -V-ATPases. All five inhibitors were dissolved in dimethyl sulfoxide (DMSO, Millipore Sigma; Cat# 41639). To keep the final concentration of DMSO to 0.1%, a 1000× stock mixture containing all five inhibitors was made. The mixture was added to pre-warmed RPMI-H to achieve the correct concentration for pre-incubation and enzymatic digestion of samples.

## Preparation of synaptosomes and myelin for antibody screening

Each mouse was deeply anesthetized with Avertin followed by cervical dislocation. Brains were isolated and submerged in 0.32 M sucrose solution (Sucrose, Millipore Sigma Cat# S0389 dissolved in milli-Q water) and allowed to cool down. HEPES (10 mM) and protease inhibitors (Millipore Sigma, Roche; Cat# 5892791001) were added to all solutions used for this preparation. The olfactory bulbs, brain stem, and cerebellum were removed, and the remaining cerebrum chopped into small pieces as when preparing for single cell suspensions. 400 µl of minced tissue was transferred to a glass tube (Cole-Parmer/Glas-Col Apparatus; Cat# 099C S73) and 1.6 ml of 0.32 M sucrose solution was added. The tissue was homogenized using a motorized tissue homogenizer (Cole-Parmer/Glas-Col (099C K54 Cat# 099C K54) with a glass pestle (Cole-Parmer/Glas-Col; Cat# 099C S62) at 3200 rpm. The homogenate was collected in a 15 ml Falcon tube and centrifuged at $1200 \times g$ for 10 min to pellet the nuclear fraction, which was discarded. The supernatant was then spun at $15,000 \times g$ for 15 min to pellet the 'crude synaptosomes'. To further enrich for synaptosomes, the pellet was resuspended in 1 ml of 0.32 M sucrose solution and layered on top of a 15 ml three-layered sucrose gradient, from top to bottom: 0.8 M (3 ml), 1.0 M (8 ml), and 1.2 M (4 ml) in a 17 ml clear thin-walled ultra-centrifuge tube (Beckman Coulter: Cat# 344061). The samples were spun at 28,000 RPM ($150,000 \times g$ at Rmax) for 2 h in an Optima XE ultracentrifuge (Beckman Coulter, Cat# A99833) with a swinging bucket rotor (SW28.1 Ti, Beckman Coulter, Cat# 342214 and 342212). Acceleration and deceleration settings of 2 and 8, respectively, were applied. Myelin was harvested from the interface between the 0.32 M and 0.8 M sucrose layers by a P1000 pipette. The synaptosomes were harvested from the interface between the 1.2 M and 1.0 M sucrose layers by gently inserting a 25 G 5/8" needle (BD precision glide Cat# 305122) through the thin wall of the tube and collecting the cloudy solution in a 5 ml syringe. The sucrose concentration was adjusted back to ~0.32 M by diluting the sample in 10 mM HEPES solution and the synaptosomes and myelin were pelleted by spinning the sample at $15,000 \times g$ for 30 min. The above protocol was developed based on established protocols[17]. Synaptosomes and myelin were resuspended in culture media to be applied in the antibody screen described below or synaptosomes were fixed for electron microscopic analysis.

## Screen for antibodies against synaptic proteins and MBP

EOC20 cells (ATCC; CAT# CRL-2469) were grown in DMEM:F12 (1:1) + L-Glutamine media (Life Technologies; Cat# 10565-042) containing 10% fetal bovine serum (FBS, Invitrogen; Cat# 26140-079), 1% Penicillin/streptomycin (Fisher Scientific; Cat# 15140122), and 20 ng/ml recombinant mouse M-CSF protein (R&D systems; Cat# 416) and expanded until the cells were 100% confluent in T150 flasks (Corning; Cat# 355001). Pelleted synaptosomes from the cerebrum of a total of six adult mice were resuspended and pooled together in 20 ml of pre-warmed culture media and added to one T150 flask containing EOC20 cells. 20 ml of culture media without synaptosomes were added to

another flask containing EOC20 cells. The cells were incubated with the synaptosomes for 60 min at 37 °C and then non-engulfed synaptosomes were removed by washing the cells three times—once in warm culture media, and twice in cold HBSS. The cells were detached using a cell scraper (Greiner Bio-One; Cat# 541070) and filtered into 15 ml tubes through a 70 µm cell strainer. The tubes were centrifuged at $500 \times g$ for 5 min and resulting pellets were resuspended in FACS buffer and transferred to 2 ml tubes. Cells were then surface-labeled (CD45, CX3CR1, CD11b) and fixed as described above with several modifications. Due to the large volume of cells, the surface labeling and fixation steps were carried out in 2 ml tubes instead of in a 96-well plate, and the staining volume was increased to 500 µl (instead of 50 µl). After fixation, each sample (fed and unfed EOC20 cells) was split into 51 wells of a 96-well plate. The cells were permeabilized and blocked with 10% goat serum, and 100 µl of each of the 51 antibody/isotype control solutions was transferred to the appropriate sample wells. Most of the antibodies were not primary conjugated. Thus, following a wash to remove unbound antibodies, cells were incubated with secondary antibodies conjugated with AF647. Before flow-cytometric acquisition, samples were also stained with DAPI to distinguish nucleated cells from debris.

This in vitro assay was additionally applied to screen for antibodies that would enable the detection of engulfed MBP as a proxy for myelin engulfment. Pelleted myelin from the cerebrum of one adult mouse was resuspended in 20 ml of pre-warmed culture media and added to one T25 flask containing EOC20 cells. 20 ml of culture media without myelin was added to another flask containing EOC20 cells. The cells were incubated with the myelin for 120 min at 37 °C and then non-engulfed myelin was removed by washing the cells three times. Cells were then detached using a cell scraper and processed as described above for synaptosome-fed cells.

## Primary microglia cultures

Single-cell suspensions from the brains of C57BL/6 mice (P18-P21) were prepared by Dounce homogenization and microglia were enriched by centrifugation in 40% Percoll (as described above). Microglia were pelleted by centrifugation at $500 \times g$, washed in HBSS, and resuspended in DMEM:F12-based media as used for culturing the EOC20 microglia. The media was changed every three days until the microglia reached 100% confluence in T25 flasks (Millipore Sigma; Cat# SIAL0639-200EA).

## Immunohistochemistry

Mice were transcardially perfused with 10 ml of cold HBSS followed by 30 ml of cold 4% PFA (16% PFA; EM grade, Electron Microscopy Science; 15710, diluted in HBSS). The brains were further post fixed in 4% PFA at 4 °C for 12 h, washed in HBSS, cryoprotected in 30% sucrose (Millipore Sigma; S0389) in HBSS for 48 h, and frozen on dry ice in a mixture containing 30% Tissue-Tek (Optimal Cutting Temperature; Sakura Finetek, 4583) and 20% sucrose in HBSS. 50 µm free-floating sections were cut using a cryostat (Leica CM 1950) and washed in HBSS. The tissue was permeabilized with 0.2% Triton X-100 (Millipore Sigma; T8787), non-specific binding was blocked by 20% goat serum (Millipore Sigma; G9023) in HBSS, and immunolabeling conducted with primary antibodies diluted in HBSS with 10% goat serum at 4 °C for 12 h. Primary antibodies included rabbit-anti-P2Y12 (Anaspec; Cat# AS-55043A, diluted 1:500 to a final concentration of 0.4 µg/ml) and rabbit-anti-Iba1 (Wako Chemicals; Cat# 019-19741, diluted 1:500 to a final concentration of 1 ug/ml). P2Y12 and Iba1 immuno-labeling were combined to provide a uniform labeling of the entire microglial morphology. Rat-anti-CD68 (clone FA-11, Bio-Rad/Serotec; Cat# MCA1957, diluted 1:250 to a final concentration of 4 µg/ml) was used to label lysosomes. Secondary antibodies included Goat-anti-rabbit IgG conjugated with AF594 (Thermo Fisher Scientific; Cat# A11037, diluted 1:500 to a final concentration of 4 µg/ml) and Goat-anti-rat IgG

conjugated with AF647 (Thermo Fisher Scientific; Cat# A11007, diluted 1:500 to a final concentration of 4 µg/ml)) were also diluted in HBSS with 10% goat serum and the tissue was incubated at RT for 2 h. The sections were then incubated with DAPI at a final concentration of 250 ng/ml (545 nM) for 30 min at RT and mounted on microscope slides (X-tra, Leica Biosystems; 3800200) in 6.25 mg/ml gelatin (from bovine skin, Millipore Sigma; G9382) diluted in $dH_2O$, and cover slipped (No. 1.5, VWR, 48393-195) in SlowFade Diamond Antifade mounting medium (Thermo Fisher Scientific; S36972).

### Immunocytochemistry and confocal microscopy imaging
Single suspensions were prepared as described above for FEAST. Instead of surface labeling the live microglia, fixed and permeabilized microglia were stained using the same combination of antibodies as used for fixed tissue. The cells were stained with DAPI, pelleted, resuspended in gelatin, and smeared on microscope slides and cover slipped in SlowFade mounting medium. Microscope slides with tissue sections and cell smears were imaged using a laser scanning confocal microscope (Zeiss LSM710, with Zen Black v2.3 software) as z-stacks, stepping 0.5 um using a 63×/1.40 Oil objective. The images were processed using Fiji/ImageJ[36].

### Reporting summary
Further information on research design is available in the Nature Portfolio Reporting Summary linked to this article.

## Data availability
The authors confirm that all relevant data are included in the paper. Source data are provided with this paper and data further supporting the findings of this study are available upon request. Source data are provided with this paper.

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

## Acknowledgements

We thank Jodene K. Moore, Ashima Agarwal, Natasha Barteneva, and Kenneth Ketman at Boston Children's Hospital Flow and Imaging Cytometry Resource (FICR) Facility for their assistance and technical support. We thank the IDDRC Cellular Imaging Core at Boston Children's Hospital (funded by NIH P50 HD105351 and S10OD016453) for technical support. We thank Joel Cuadrado, Alanna Carey, and Arnaud Frouin as well as the entire staff at ARCH for genotyping and husbandry. We thank Aniqa Tasnim, Rui Peixoto, and Etienne Herzog for their assistance in optimizing the protocol for preparing synaptosomes. We thank Lauren Mifflin for her help in optimizing the microglia enrichment protocol. We thank Theodore Fisher for his efforts on light sheet imaging. We thank Yvanka de Soysa, Ravi L. Rungta, and Timothy R. Hammond for feedback on earlier versions of the manuscript. This work was funded by grants from: National Institutes of Health, Silvio O. Conte Centers for Basic Neuroscience or Translational Mental Health Research (Sponsor Grant# 5P50MH112491-04) (B.S.). National Institute of Health, National Institute of Neurological Disorders and Stroke (Sponsor Grant # 5R01NS092578-05) (B.S.). Howard Hughes Medical Institute (B.S.) Stanley Center for Psychiatric Research, Broad Institute (B.S.) Dr. Miriam and Sheldon G. Adelson Medical Research Foundation (L.I.B.) Medical Technology Enterprise Consortium (L.I.B.) Lundbeck Foundation International Postdoc Fellowships (L.D-O.) Kirschstein-NRSA training award 5 T32 AG 222-30 (H.J.B.) HD Human Biology fellowship (D.K.W.) HCRP research award and Herchel Smith fellowship (I.D.)

## Author contributions

L.D-O. and A.J.W. carried out the majority of experiments and analyses with support from co-authors. Q.F. and L.X. performed optic nerve crushes. H.J.B. performed the CPZ experiments. A.C.W. performed the LPC experiments. D.K.W. helped optimized the protocol for preparing synaptosomes and facilitated the acquisition of electron microscopy images. I.D. did immunohistochemistry and assisted with single-cell preparations and EOC20 cultures. L.D-O., A.J.W., L.I.B., and B.S. designed the study and wrote the manuscript with contributions from all authors.

## Competing interests

The authors declare no competing interests.
