## [Peer Review File · Nature Communications]

REVIEWER COMMENTS

Reviewer #1 (Remarks to the Author):

The manuscript by Dissing-Olesen describes a FASC-based method to analyze and quantify cellular material that microglia had taken up in vivo. This new protocol builds on earlier published protocols and convincingly addresses some of the shortcomings of those earlier studies. Especially the thorough assessment of the different isolation methods and the provided list of suitable antibodies for synaptic markers represent an improvement that will enable further studies. As pointed out by the authors, microglia phagocytosis is important in brain homeostasis and disease. This method may enable a better understanding of how microglia phagocytosis is controlled and how it impacts microglia physiology.

The title and the name of this protocol suggests that it is possible to specifically analyse engulfment of synaptic material by microglia. But this is not correct, which also is discussed by the authors, and therefore potentially misleading. The optic nerve crush and the demyelination models both create cell loss and one might assume that the microglia are cleaning up corpses or non-functional neurons in response rather than specifically taking up synaptic material only. To assess whether the here provided protocol does allow for analysis of microglia synapse interactions more physiological models without cell loss should be used.

Reviewer #2 (Remarks to the Author):

In this manuscript, Dissing-Olesen and colleagues present a method for assessing mouse microglia and brain border-associated macrophage (BAM) engulfment of synaptic and myelin debris using flow cytometry. It provides new, detailed protocols for both fresh and fixed tissue with nuance and careful controls to assess microglia debris engulfment in vivo and is of great interest given the need to assess microglia engulfment of brain components in health and in disease. It could be of wide interest to the readership of the journal because of the potential applicability to other prey and as a framework for developing similar systems for other tissue-resident microglia. There are two major and few minor concerns that if addressed would improve the impact of this work.

Major concerns:

This technique provides a feasible method for identifying engulfment of fluorescently labeled particles by flow cytometry. They show the contrasts of two common brain macrophage isolation methods and land on isolation from fixed tissue as the least “contaminated” approach and with use of molecular inhibitors. In light of this and the possibility that these different isolation methods might isolate different

populations of cells, the major benchmark is how these MFIs compare with in vivo methods. Recommend authors firm up their myelin engulfment story with histological analysis of BAM versus microglia engulfment as an orthogonal method.

As this work provides a semi-quantitative method for assessing microglia engulfment, and a future application could include something like high throughput screens, some measurement of a dynamic change would enable better extrapolation - for example, why are the time points for the assays selected? Evaluating engulfment of either myelin or synaptic input at one or two additional time points after the insult would better establish this method as robust and adaptable.

Minor Concerns:

Would enable the more facile use of these great techniques if the supplement contains a “protocols” style step by step method

What was the thought process behind the timepoints selected for analyses (related to point 2 above)? Recommend highlighting this

The idea of sniffer cells is excellent. For clarity, suggest making it more apparent as the concept is introduced that one would not expect to find SYN immunostaining in SYN KO cells and highlight that this is what you see in Fig 4d and 4e. The clause at the end of the first paragraph page 10 made the idea more difficult than necessary to grasp.

Also for clarity in supplemental table would make apparent which antibodies/clones were the ones that passed the screen (I believe the first listed ones but warrants special denotation)

Reviewer #3 (Remarks to the Author):

Dissing-Olesen and Walker et al described a high-throughput flow cytometry-based method FEAST for quantitative assessment of microglia engulfment of synaptic terminals and myeline at single-cell resolution. This is a very demanding technique for the field and can be expanded as the authors have mentioned to study other targets such as quantifying the uptake of protein aggregates. The FEAST technology has largely expanded the available toolbox for assessment of microglial phagocytosis, which is one of the key function of the immune cells in the brain.

The study was well designed and controlled, the methods are sound and the results support the authors' conclusions. The paper is very well written. I believe this work will considerably benefit the field.

Before fully support its publication. I have three minor points that wish the authors to address:

1. The authors have established FEAST as a new method for quantification of microglial cell engulfment, and the perfusion-fixed FEAST approach seems the most reliable method. It is valuable to validate/confirm the results obtained from FEAST e.g. Figure 5b through an orthogonal methods for instance classical immunostaining to see how much discrepancy the different methods could be, and this will infer the reliability of the FEAST technology.

2. In Figure 2, the authors tested microglial engulfment of different FPs, the results are interesting and reasonable. Although the expression level and distribution of these different FPs in the tested brain region should be comparable, presenting the data would exclude the possibility that the observed difference of the FPs in microglia were not due to the variable expression or distribution of these FPs.

3. In Figure 1, the results indicated and the authors also concluded that false-positive signal through ex vivo engulfment of neuronal material is negligible using the Dounce homogenization method, however, using the same isolation method, in Extended Figure 2, there is high false-positive signal. How would the authors explain? A discussion on it may facilitate the understanding of the technology in different paradigms better.

Reviewer #4 (Remarks to the Author):

Dissing-Olesen, Walker and colleagues describe an elegant new approach to studying engulfment that they term FEAST (flow cytometric engulfment assay for synaptic terminals). They find that flow cytometry can be used to evaluate engulfment while demonstrating the various caveats and challenges with well-thought-out control experiments. I think FEAST will be of broad interest to those studying engulfment. In the end, they find three valuable protocols to define the engulfment of synaptic terminals with fluorescent proteins, intracellular antibodies with living cells, and intracellular antibodies with fixed cells. They provide several options of antibodies so those studying other synaptic targets or myelin can use flow cytometry to measure engulfment. I thought using FEAST to examine the engulfment of fixed tissue was especially inspired, and I hope to try this in my lab. One small caveat of the name is that it infers synaptic targets, but as the authors articulate in the discussion, this approach is more powerful than just measuring synapses. That said, FEAST is a sticky acronym.

I thought this manuscript was very well composed and the control experiments well planned, so my comments are only very minor. Some reviewers might ask the authors to use FEAST with more outcomes to show its versatility, but I believe this excellent work should not be held back.

Minor points

Please include representative gating strategy for figures 1 and/or 2

Please include more information on the flow plots based on recommendations by the minimal information of flow cytometry experiment (18752282). For example, ideally, you add the antigen, fluorochrome, and optical filter. Please keep the numbers used for scaling on X/Y axis. Also include the number of events in all of the flow plots.

For the primary microglia protocol, it was unclear how microglia were purified. This should be stated if this is just a mixed glial culture.

Why do WT microglia take up Syn1 ex vivo, but sniffer cells do not (Fig 4d)?

Were mice ever pooled in these in vivo experiments? If so, please include animal numbers in the figure legends. If animals are never pooled, then the cell number of the flow plot will give the reader a sense of how many cells can be isolated per animal, which I personally would find very helpful.

lab worthy protocols for the three major protocols outlined in extended data 6 would help to facilitate adaptation for labs wanting to use these approach

REVIEWER COMMENTS

Reviewer #1 (Remarks to the Author):

The manuscript by Dissing-Olesen describes a FASC-based method to analyze and quantify cellular material that microglia had taken up in vivo. This new protocol builds on earlier published protocols and convincingly addresses some of the shortcomings of those earlier studies. Especially the thorough assessment of the different isolation methods and the provided list of suitable antibodies for synaptic markers represent an improvement that will enable further studies. As pointed out by the authors, microglia phagocytosis is important in brain homeostasis and disease. This method may enable a better understanding of how microglia phagocytosis is controlled and how it impacts microglia physiology. The title and the name of this protocol suggests that it is possible to specifically analyse engulfment of synaptic material by microglia. But this is not correct, which also is discussed by the authors, and therefore potentially misleading. The optic nerve crush and the demyelination models both create cell loss and one might assume that the microglia are cleaning up corpses or non-functional neurons in response rather than specifically taking up synaptic material only. To assess whether the here provided protocol does allow for analysis of microglia synapse interactions more physiological models without cell loss should be used.

We thank reviewer 1 for their positive assessment of our manuscript and helpful comments. We agree that the acronym FEAST, where ST stands for “Synaptic Terminals” is too specific and potentially misleading given that this method can be applied to examine engulfed myelin and theoretically many other engulfed proteins. This point was raised by reviewer 4 as well. We have now changed the name of the method to broaden the scope but maintain the acronym:

FEAST: Flow cytometry Engulfment Assay for Specific Target proteins

Regarding the ability of FEAST to specifically detect engulfment of synaptic material, we believe that this depends on the context in which it is being employed. The ONC paradigm we have used evokes Wallerian degeneration of retinal ganglion cell (RGC) axons and the concomitant loss of non-functional synaptic terminals specifically in the brain regions that were isolated by micro-dissection (LGN & SC). In this context, the microglia we analyze were only exposed to degenerating synaptic terminals (and some axonal material), but not the RGC corpses which exist in the retina (PMC6054657). While this is not physiological synaptic pruning, it is still a paradigm in which engulfment of synaptic material derived from synaptic terminals can be assessed in an environment and time frame during which loss of cells is not taking place. In less anatomically restricted areas or other paradigms, we agree that it will be difficult to distinguish specific uptake of synaptic terminals from more general engulfment of synaptic proteins from dying cells. This is a general limitation of all techniques in the field that are currently used to assess the uptake of synaptic material.

We agree that there will be corpses of oligodendrocytes in our models of demyelination as these models (LPC and CPZ) do not allow a physical separation of the region that is being demyelinated and the region containing oligodendrocyte corpses. Thus, we are only using MBP as a proxy for the engulfment of myelin and cannot comment on from where this myelin was acquired. Models that will allow us to distinguish these possibilities are beyond the scope of the current methods paper but will be a powerful application of FEAST that we hope others in the field pursue.

Reviewer #2 (Remarks to the Author):

In this manuscript, Dissing-Olesen and colleagues present a method for assessing mouse microglia and brain border-associated macrophage (BAM) engulfment of synaptic and myelin debris using flow cytometry. It provides new, detailed protocols for both fresh and fixed tissue with nuance and careful controls to assess microglia debris engulfment in vivo and is of great interest given the need to assess microglia engulfment of brain components in health and in disease. It could be of wide interest to the readership of the journal because of the potential applicability to other prey and as a framework for developing similar systems for

other tissue-resident microglia. There are two major and few minor concerns that if addressed would improve the impact of this work.

We thank Reviewer #2 for their positive comments and helpful suggestions. We have addressed the major and minor concerns below.

Major concerns:

This technique provides a feasible method for identifying engulfment of fluorescently labeled particles by flow cytometry. They show the contrasts of two common brain macrophage isolation methods and land on isolation from fixed tissue as the least “contaminated” approach and with use of molecular inhibitors. In light of this and the possibility that these different isolation methods might isolate different populations of cells; the major benchmark is how these MFIs compare with *in vivo* methods. Recommend authors firm up their myelin engulfment story with histological analysis of BAM versus microglia engulfment as an orthogonal method.

We agree that it is necessary to provide evidence that our FEAST protocol for measuring *in vivo* engulfment can broadly recapitulate phenotypes previously identified with imaging-based approaches. In both paradigms we have studied - engulfment of myelin proteins after demyelination and of synapse-associated labels after ONC - we can qualitatively replicate previously published findings.

Others have assessed the engulfment of myelin using the classic IHC and microscopy-based analyses of fixed tissue (e.g., PMC7101330 and PMC7498497), which prompted us to examine this phenotype by FEAST as a proof-of-principle. As previously described, we also detect an increase in myelin engulfment by microglia in demyelinated mice compared to controls (Figure 7). This qualitative comparison between FEAST and IHC-based methods also holds true in our assessment of engulfment of fluorescent CTB after ONC, where we were able to visualize CTB-AF488 inside of microglia *in situ* by IHC, as well as by ICC of acutely isolated microglia, and then to quantify this engulfment by flow cytometry (Figure 1). The relative increase in engulfment at different time points post ONC also matches the IHC engulfment data quantified in Norris et al. 2018 (PMC6028515, Figure 1M). Just as they reported, we also observed a greater increase in engulfment in ONC compared to controls at 3 days post ONC vs 8 days post ONC by FEAST (Reviewer Figure 1).

However, a direct comparison between the MFI of engulfed proteins detected by FEAST and the metrics of engulfment commonly assessed by IHC/microscopy is not feasible because they aim to quantify different features: FEAST provides quantification of the MFI of engulfed material, whereas IHC-based methods examine the volume of engulfed signal in the microglia or the number of substrate inclusions within lysosomes. Thus, we believe that our ability to detect an increase in engulfed synaptic and myelin material in models where this has also been demonstrated by IHC is as far as the direct comparison between FEAST and IHC-based engulfment methods can currently be taken.

In general, we believe that the different methodologies (IHC vs FEAST) can be viewed as orthogonal in the sense of addressing questions about *in vivo* engulfment; however, they do so at very different scales. For example, IHC offers great spatial resolution to investigate anatomically distinct populations of cells (e.g., cells in particular brain sub-regions or near pathological features like plaques). The anatomical specificity of FEAST is less precise as it depends on micro-dissections of the region(s) of interest. In contrast, the major strength of FEAST compared to IHC is that allows rapid interrogation of a much higher number of cells and replicates, providing greater power to discriminate differences. We hope that the field will adopt FEAST as an additional powerful tool to investigate engulfment but assert that the information gained from FEAST is not directly comparable to that generated by IHC.

As this work provides a semi-quantitative method for assessing microglia engulfment, and a future application could include something like high throughput screens, some measurement of a dynamic change would enable better extrapolation - for example, why are the time points for the assays selected? Evaluating engulfment of either myelin or synaptic input at one or two additional time points after the insult would better establish this method as robust and adaptable.

We agree that FEAST will be a powerful tool for many applications including high throughput screens, and that in such applications the dynamic range and time window during which the assay is conducted will be critical. We indeed selected the timepoints for the ONC and demyelination paradigms based on previous literature and pilot experiments in our own lab.

For ONC:

Norris et al. 2018 (PMC6028515) nicely demonstrate by IHC that microglial engulfment of CTB-AF647 peaks at 3 days post ONC in the LGN. In pilot experiments using FEAST, we compared engulfment of CTB at 3 days post ONC and 8 days post ONC, confirming that engulfment was highest at 3 days (Reviewer Figure 1). We therefore continued with analyses at 3 days post ONC for subsequent experiments using the ONC paradigm.

Reviewer Figure 1:

a) Schematic of CTB-AF594 labeled RGC and micro-dissected LGNs & SCs. RGCs in each eye were labeled by intravitreal injections of CTB-594 prior to ONC. Microglia were isolated from micro-dissected and pooled LGNs and SCs for quantification of engulfed AF594 by flow cytometry 3- and 8-days post ONC. **b)** Representative flow cytometry plots of microglia indicate that a greater proportion of microglia are positive for AF594 at 3 days after ONC compared to 8 days after ONC. Quantification of the MFI for AF594 in microglia also indicates a much greater delta between ONC and no crush when engulfment was examined 3 days post ONC compared to 8 days post ONC. n=2 mice per condition.

For CPZ demyelination:

Previous literature shows that oligodendrocyte loss occurs as early as 2 weeks post cuprizone administration and continues to 5-6 weeks post cuprizone administration (e.g., PMC7329620). In pilot experiments using FEAST, we compared engulfment of MBP at 2 weeks, 3 weeks, and 5 weeks post cuprizone administration (Reviewer Figure 2) and settled on analyzing 3 weeks after initiation of CPZ treatment.

Reviewer Figure 2:

a) Representative flow cytometry plots for microglia after 2, 3, and 5 weeks of treatment with CPZ indicate that a greater proportion of microglia are positive for MBP at 3 weeks compared to 2 and 5 weeks of treatment. b) Microglia were isolated from micro-dissected and pooled Cx and CC as indicated with the dotted orange line. Note, for the experiments presented in the manuscript, Fig 7. Cx and CC were analyzed separately. c) Quantification of the MFI for engulfed MBP in microglia indicates the greatest delta between CPZ and control can be observed after 3 weeks of CPZ treatment. n=3-4 mice per time point.

Minor Concerns:

Would enable the more facile use of these great techniques if the supplement contains a “protocols” style step by step method:

We have now included a step-by-step protocol for each of the three FEAST approaches in Extended Data Figs 7, 8, and 9. This is complemented by Extended Data Table 1, 2, and 3, which describe antibodies used for detection of synaptic proteins and myelin proteins, and antibodies used to distinguish microglia from BAMs for the different FEAST applications. We hope this will enable facile implementation of FEAST by many in the field.

What was the thought process behind the timepoints selected for analyses (related to point 2 above)? Recommend highlighting this.

We have now addressed this as part of our response to point 2.

The idea of sniffer cells is excellent. For clarity, suggest making it more apparent as the concept is introduced that one would not expect to find SYN immunostaining in SYN KO cells and highlight that this is what you see in Fig 4d and 4e. The clause at the end of the first paragraph page 10 made the idea more difficult than necessary to grasp.

Thank you – we believe that the “sniffer cell” approach was critical to evaluating the validity of our FEAST protocols. We have now clarified the concept and rationale in the text as suggested. Line 3-6, page 10: “GFP⁺ microglia and BAMs (referred to as “sniffer cells”) are derived from SYN1-KO brains, and thus will only contain SYN1 material if they have acquired it aberrantly ex vivo from the WT brain, they were mixed with during tissue dissociation or subsequent steps of FEAST (Fig 4a and c).”

Also for clarity in supplemental table would make apparent which antibodies/clones were the ones that passed the screen (I believe the first listed ones but warrants special denotation).

We have now indicated the synaptic and myelin antibodies that we consider applicable to FEAST by highlighting them with yellow and marking them with an asterisk in Extended Data Table 1.

Reviewer #3 (Remarks to the Author):

Dissing-Olesen and Walker et al described a high-throughput flow cytometry-based method FEAST for quantitative assessment of microglia engulfment of synaptic terminals and myeline at single-cell resolution. This is a very demanding technique for the field and can be expanded as the authors have mentioned to study other targets such as quantifying the uptake of protein aggregates. The FEAST technology has largely expanded the available toolbox for assessment of microglial phagocytosis, which is one of the key function of the immune cells in the brain.

The study was well designed and controlled, the methods are sound and the results support the authors' conclusions. The paper is very well written. I believe this work will considerably benefit the field.

We thank Reviewer #3 for their positive assessment of our manuscript and helpful comments and have addressed their outstanding concerns below.

Before fully support its publication. I have three minor points that wish the authors to address:

1. The authors have established FEAST as a new method for the quantification of microglial cell engulfment, and the perfusion-fixed FEAST approach seems the most reliable method. It is valuable to validate/confirm the results obtained from FEAST e.g. Figure 5b through an orthogonal methods for instance classical immunostaining to see how much discrepancy the different methods could be, and this will infer the reliability of the FEAST technology.

We agree that it is necessary to provide evidence that our FEAST protocol for measuring *in vivo* engulfment can broadly recapitulate phenotypes previously identified with imaging-based approaches. Indeed, previous assessments of engulfment using classical tissue immunostaining were an important reason why we selected the ONC paradigm for establishing and validating FEAST.

Norris et al. 2018 (PMC6028515) had already examined engulfment in this paradigm by IHC. We qualitatively reproduced their results, demonstrating CTB-AF488 inclusions in microglia after ONC (Fig 1a). We next performed ICC on microglia isolated from the LGN/SC after ONC to confirm that these inclusions remained inside of the cells (Fig 1b) and then quantified CTB-AF488 and SNAP-25 engulfment by flow cytometry (Fig 1 d-e, Reviewer Fig 1, and Fig 6b-c), demonstrating an increase in engulfed substrates after ONC compared to no crush. We believe that this qualitative comparison (e.g., detection of a change in engulfment from baseline) is as far as the comparison between FEAST and IHC-based engulfment methods can be taken currently, as they rely on very different readouts: FEAST provides quantification of the MFI of engulfed material, whereas IHC-based methods examine the volume of engulfed signal in the microglia or the number of substrate inclusions within lysosomes.

In general, we believe that the different methodologies (IHC vs FEAST) can be viewed as orthogonal in the sense of addressing questions about *in vivo* engulfment; however, they do so at very different scales. For example, IHC offers great spatial resolution to investigate anatomically distinct populations of cells (e.g., cells in particular brain sub-regions or near pathological features like plaques). The anatomical specificity of FEAST is less precise as it depends on micro-dissections of the region(s) of interest. On the other hand, the major strength of FEAST compared to IHC is that it allows rapid interrogation of a much higher number of cells and replicates, providing greater power to discriminate differences. We hope that the field will adopt FEAST as an additional powerful tool to investigate microglia and macrophage biology but assert that the information gained from FEAST is not directly comparable to that generated by IHC.

2. In Figure 2, the authors tested microglial engulfment of different FPs, the results are interesting and reasonable. Although the expression level and distribution of these different FPs in the tested brain region should be comparable, presenting the data would exclude the possibility that the observed difference of the FPs in microglia were not due to the variable expression or distribution of these FPs.

This is a good point. Initially, our goal was to fluorescently label pre- vs post-synaptic terminals, or inhibitory versus excitatory synapses, with different FPs (e.g., using viral strategies or synaptic tagged reporter mice) and then apply FEAST to address questions about the engulfment of these different synaptic elements in the same mouse. However, we soon realized that this would be challenging given potential differences in pKa, lysosomal stability, toxicity, fluorescent intensity, and abundance of spectrally distinguishable FPs that we could express. Instead, we decided to examine whether we could sensitively detect engulfment of four common FPs that were available as lox-stop reporter mice and thus could be expressed by the same Cre driver (CHX10cre).

During the experiments, we ensured that each of the four different FPs was expressed by the RGCs and resulted in a bright fluorescent signal in the LGNs and SCs, which allowed for micro-dissection of these regions facilitated by a fluorescent dissection microscope. An example of the expression for EYFP is displayed in Reviewer Figure 3. However, as pointed out by the reviewer, these proteins are likely expressed at different levels and do have notably different reported brightness (both of which will affect the MFI of their engulfed signal). This is the reason that we treated engulfment data from each FP independently and display the results for each FP on a separate graph to avoid cross-FP comparison of MFIs. Additionally, the statistical analyses are restricted to no crush vs ONC for each individual FP.

As stated in both the Results and Discussion sections of the manuscript, our goal with these assessments is to provide some guidance for the selection of FPs and to encourage readers to carefully consider which FPs they select for their experimental design. Further screening of FPs with different pKa and lysosomal stability will likely reveal better candidates that can be used in combination with FEAST.

Reviewer Figure 3:

Example of FP expression when the CHX10cre line is crossed with a FP reporter line. In the eye and the brain of mice from CHX10cre x Isl:EYFP crosses, EYFP is observed in the retina, the optic nerve, LGNs, and SCs, allowing for micro-dissection of the LGNs and SCs.

3. In Figure 1, the results indicated and the authors also concluded that false-positive signal through ex vivo engulfment of neuronal material is negligible using the Dounce homogenization method, however, using the same isolation method, in Extended Figure 2, there is high false-positive signal. How would the authors explain? A discussion on it may facilitate the understanding of the technology in different paradigms better.

We appreciate this insightful point and understand the confusion that these two seemingly inconsistent results may cause. In Figure 1 (and Figure 2) we assess engulfment of fluorescent neuron-derived material in microglia that are isolated from the brain by Dounce homogenization but kept alive during processing and analysis (no fixation/permeabilization/intracellular staining steps). In this context, we consistently observe minimal *ex vivo* contamination of microglia with exogenous fluorescent material. In Extended Data 2, we assess immunostaining for SYN-1 associated with cells that are fixed and permeabilized after isolation from the brain. Here, we see abundant false-positive contamination in cells isolated by Dounce homogenization. We hypothesize that false positive signals in this context is generated by the adhesion of SYN-1⁺ debris specifically to fixed/permeabilized cells. Why this contamination does not happen in cells that were isolated from brains dissociated with collagenase IV is unclear but may have to do with differences in the amount of SYN-1⁺ debris present in the samples after gradient purification and/or with the nature of debris (e.g., size, association with other structures) generated by the different preparations. We have now clarified this rationale in the text. Line 15-20, page 19: *“The “sniffer cell” approach allowed us to determine that microglia acquire substantial false-positive SYN1 signal when dissociated by Dounce homogenization and then fixed, permeabilized, and stained intracellularly for engulfed material. This data contrasts to what we observed for microglia isolated by Dounce homogenization and then kept alive to analyze engulfment of pre-labeled neuronal material, suggesting that fixed/permeabilized cells may be more vulnerable to adherence of false-positive debris in certain contexts.”*

Reviewer #4 (Remarks to the Author):

Dissing-Olesen, Walker and colleagues describe an elegant new approach to studying engulfment that they term FEAST (flow cytometric engulfment assay for synaptic terminals). They find that flow cytometry can be used to evaluate engulfment while demonstrating the various caveats and challenges with well-thought-out control experiments. I think FEAST will be of broad interest to those studying engulfment. In the end, they find three valuable protocols to define the engulfment of synaptic terminals with fluorescent proteins, intracellular antibodies with living cells, and intracellular antibodies with fixed cells. They provide several options of antibodies so those studying other synaptic targets or myelin can use flow cytometry to measure engulfment. I thought using FEAST to examine the engulfment of fixed tissue was especially inspired, and I hope to try this in my lab. One small caveat of the name is that it infers synaptic targets, but as the authors articulate in the discussion, this approach is more powerful than just measuring synapses. That said, FEAST is a sticky acronym. I thought this manuscript was very well composed and the control experiments well planned, so my comments are only very minor. Some reviewers might ask the authors to use FEAST with more outcomes to show its versatility, but I believe this excellent work should not be held back.

We thank Reviewer #4 for their enthusiastic evaluation of our manuscript, and for their helpful minor comments. We agree that the title FEAST, where ST stands for synaptic terminals, might not reflect the power of the method. We have consequently altered the name to reflect the broader applicability while maintaining the acronym: **FEAST: Flow cytometry Engulfment Assay for Specific Target** proteins.

Minor points

Please include representative gating strategy for figures 1 and/or 2

We have now included these representative gating strategies in Extended Data 13.

Please include more information on the flow plots based on recommendations by the minimal information of flow cytometry experiment (18752282). For example, ideally, you add the antigen, fluorochrome, and optical filter. Please keep the numbers used for scaling on X/Y axis. Also include the number of events in all of the flow plots.

We have now included the antigen, fluorochrome, X/Y scaling, and cell number on all flow cytometry plots. For further clarity, we have included the requested information about cytometer configuration/band pass filters as a separate table (Extended Data Table 4).

For the primary microglia protocol, it was unclear how microglia were purified. This should be stated if this is just a mixed glial culture.

We appreciate this question and have now included details about microglia isolation for culture in the figure legend for Extended Data 1. Briefly, primary microglia were isolated by cold Dounce homogenization followed by 40% Percoll enrichment. This is the same as the isolation protocol for FEAST on live microglia, and the details are described in this part of the Methods section and referenced in the section about the isolation of primary microglia for culture.

Why do WT microglia take up Syn1 *ex vivo*, but sniffer cells do not (Fig 4d)?

This is a very good question and a potential limitation of the “sniffer cell” approach. We hypothesize that the SYN-1 signal in the WT microglia is due to *ex vivo* engulfment of synaptic material that is physically adjacent to the microglia in brain tissue chunks while the tissue is being digested. Thus, “sniffer” microglia are mostly exposed to an environment devoid of SYN-1 material (inside a SYN-1 KO brain tissue chunk) during the digest and do not take up SYN-1 during this step. The picture may be more complicated for “sniffer” BAMs, which sit in the perivascular and leptomeningeal spaces and thus are likely more exposed to free-floating SYN-1⁺ material that is released during the digest. Beginning at the trituration step, both “sniffer” microglia and BAMs will be exposed to abundant SYN-1⁺ debris, making this control particularly relevant to assess adhesion or uptake of debris during the rest of the FEAST protocol. This apparent limitation of “sniffer cells”, especially for assessing false positive signals acquired by microglia during the digest, is one reason that we developed the *in vivo* fixation approach. This step eliminates the possibility that microglia are actively engulfing material *ex vivo*.

Were mice ever pooled in these *in vivo* experiments? If so, please include animal numbers in the figure legends. If animals are never pooled, then the cell number of the flow plot will give the reader a sense of how many cells can be isolated per animal, which I personally would find very helpful.

We have now included information about cell numbers on flow plots in the manuscript. We did not pool mice for any experiments included here, as the FEAST protocols produce a sufficient yield of microglia for analysis from individual mice.

lab worthy protocols for the three major protocols outlined in extended data 6 would help to facilitate adaptation for labs wanting to use these approach

We appreciate this point and have now included detailed protocols for each of the three FEAST methods as Extended Data 7, 8, and 9 (Extended Data Fig 7: FEAST on live cells isolated by cold Dounce homogenization, Extended Data Fig 8: FEAST on cells isolated by enzymatic digestion with inhibitor cocktail, and Extended Data Fig 9: FEAST on cells harvested from perfusion fixed tissue). We hope this will facilitate easy implementation of FEAST by many labs in the field.

REVIEWERS' COMMENTS

Reviewer #1 (Remarks to the Author):

The authors have addressed all the comments that I had in the first review round. Good job, the manuscript now reads very nicely, this method will be of importance for the field.

Reviewer #2 (Remarks to the Author):

The authors have sufficiently answered this reviewer concerns. The microscopy images provided in the rebuttal (And the manuscript, of course) are beautiful!

Reviewer #3 (Remarks to the Author):

All my concerns have been addressed. I am happy to support its publication in Nature Communications now.

Reviewer #4 (Remarks to the Author):

The authors have addressed my concerns
Great work!!

Point-by-point response to reviewer's comments

Reviewer #1 (Remarks to the Author):

The authors have addressed all the comments that I had in the first review round. Good job, the manuscript now reads very nicely, this method will be of importance for the field.

We thank reviewer 1 for their positive comments.

Reviewer #2 (Remarks to the Author):

The authors have sufficiently answered this reviewer concerns. The microscopy images provided in the rebuttal (And the manuscript, of course) are beautiful!

We are pleased to hear that reviewer 2 found our answers satisfactory and we are grateful for the compliment on the microscopy images.

Reviewer #3 (Remarks to the Author):

All my concerns have been addressed. I am happy to support its publication in Nature Communications now.

We thank reviewer 3 for their support and we are happy that we succeeded in addressing all their concerns

Reviewer #4 (Remarks to the Author):

The authors have addressed my concerns

Great work!!

We thank reviewer 4 for their enthusiasm for our work.